# A novel approach to characterize the variability in mass-Dimension relationships: results from MC3E

Joseph A. Finlon[1], Greg M. McFarquhar[2,3], Stephen W. Nesbitt[1], Robert M. Rauber[1], Hugh Morrison[4], Wei Wu[2], and Pengfei Zhang[2,5]

[1]Department of Atmospheric Sciences, University of Illinois at Urbana-Champaign, Urbana, IL 61801, United States
[2]Cooperative Institute for Mesoscale Meteorological Studies, University of Oklahoma, Norman, OK 73072, United States
[3]School of Meteorology, University of Oklahoma, Norman, OK 73072, United States
[4]National Center for Atmospheric Research, Boulder, CO 80301, United States
[5]NOAA/National Severe Storms Laboratory, Norman, OK 73072, United States

**Correspondence:** Greg M. McFarquhar (mcfarq@ou.edu)

**Abstract.** Mass-dimension ($m$-$D$) relationships determining bulk microphysical properties such as total water content (*TWC*) and radar reflectivity factor (*Z*) from particle size distributions are used in both numerical models and remote sensing retrievals. The $a$ and $b$ coefficients representing $m = aD^b$ relationships, however, can vary significantly depending on meteorological conditions, particle habits, definition of particle maximum dimension, the probes used to obtain the data, techniques used to process the cloud probe data, and other unknown reasons. Thus, considering a range of *a,b* coefficients may be more applicable for use in numerical models and remote sensing retrievals. Microphysical data collected by two-dimensional optical array probes (OAPs) installed on the University of North Dakota Citation aircraft during the Mid-latitude Continental Convective Clouds Experiment (MC3E) were used in conjunction with *TWC* data from a Nevzorov probe and ground-based S-band radar data to determine $a$ and $b$ using a technique that minimizes the chi-square difference between *TWC* and *Z* derived from the OAPs and that directly measured by a *TWC* probe and radar. All $a$ and $b$ within a specified tolerance were regarded as equally plausible solutions. Of the 16 near-constant temperature flight legs analyzed during the 25 April, 20 May, and 23 May 2011 events, the derived surfaces of solutions on the first two days where the aircraft sampled stratiform cloud had a larger range in $a$ and $b$ for lower temperature environments that correspond to less variability in *N(D)*, *TWC*, and *Z* for a flight leg. Because different regions of the storm were sampled on 23 May, differences in the variability of *N(D)*, *TWC*, and *Z* influenced the distribution of chi-square values in (*a,b*) phase space and the specified tolerance in a way that yielded 2.8 times fewer plausible solutions compared to the flight legs on the other dates. These findings show the importance of representing the variability in *a,b* coefficients for numerical modeling and remote sensing studies rather than assuming fixed values, as well as the need to further explore how these surfaces depend on environmental conditions in clouds containing ice hydrometeors.

## 1 Introduction

Mass-dimension (*m-D*) relations are required to link bulk microphysical properties, such as total water content (*TWC*) and forward model radar reflectivity factor (*Z*), to ice crystal particle size distributions (PSDs). These relations are extensively

assumed in both numerical models and remote sensing retrievals and relate a particle's mass ($m$) to its size, typically defined by its maximum dimension projected onto a 2-D plane ($D$), by means of a power law in the form $m = aD^b$. Past studies have suggested the exponent $b$ is related to the exponent in surface area-dimension relationships (Fontaine et al., 2014) or to a particle's fractal dimension (Schmitt and Heymsfield, 2010). The prefactor $a$ has some dependence on $b$ and on the particle density.

Prior $m$-$D$ relationships have been determined using cloud probe data obtained in a variety of environmental conditions. Figure 1a shows how $m$-$D$ coefficients derived from previous studies vary depending on the types of clouds sampled. A full list of these $m$-$D$ coefficients and their corresponding references is available as a supplement. Coefficients derived using data over mountainous terrain (e.g., Nakaya and Terada, 1935; Locatelli and Hobbs, 1974), cirrus clouds (e.g., Heymsfield, 1972; Hogan et al., 2000), convective clouds (e.g., Liu and Curry, 2000; Cazenave et al., 2016; Leroy et al., 2016), regions of large scale ascent (e.g., Szyrmer and Zawadzki, 2010), and computer-generated shapes (e.g., Matrosov, 2007; Olson et al., 2016) are shown. A total of 119 relations are shown in Fig. 1. The range of $a$ in Fig. 1a spans five orders of magnitude, with variations in $a$ spanning 3 orders of magnitude or more even for measurements obtained in the same cloud type. The exponent $b$ ranges between one and three within the same environments. The relations in Fig. 1 were derived using data collected by different types and versions of cloud probes, using different algorithms to process the data. McFarquhar et al. (2017) have shown that it can be difficult to disentangle the dependence of derived microphysical parameters on environmental conditions from the dependence on the probes used to collect and the methods to process the data.

Figure 1b shows that $m$-$D$ coefficients also vary depending on the technique used to derive the $m$-$D$ relations. In some studies the maximum dimension of frozen hydrometeors was recorded before the crystal was melted and the single particle mass subsequently measured (Magono and Nakamura, 1965; Zikmunda and Vali, 1972; Mitchell et al., 1990), whereas other studies used measurements of either bulk mass measured by an evaporation probe (Heymsfield et al., 2002; Cotton et al., 2013; Xu and Mace, 2017) or bulk Z observed by a collocated radar measurement (McFarquhar et al., 2007a; Maahn et al., 2015) in combination with in situ measured PSDs. Further, Wu and McFarquhar (2016) showed inconsistencies in how $D$ is defined (Mitchell and Arnott, 1994; Brown and Francis, 1995; McFarquhar and Heymsfield, 1996; Heymsfield et al., 2013; Lawson et al., 2015; Korolev and Field, 2015) can also impact $m$-$D$ relations. For example, they noted ice water content (IWC) values derived using various definitions of $D$ ranged between 60 and 160% of the IWC derived using a smallest enclosing circle to define $D$.

Remote sensing retrieval schemes and model microphysical parameterization schemes are sensitive to the choice of $m$-$D$ relationship. For example, Delanoë and Hogan (2010) showed that differences in the mean extinction, IWC, and effective radius retrieved from spaceborne remote sensors were 28, 9, and 30%, respectively, depending on whether $m$-$D$ relations of spherical aggregates (Brown and Francis, 1995, hereafter BF95) or bullet rosettes (Mitchell, 1996) were used. McCumber et al. (1991) showed time series of modeled precipitation rate with differences of 20 to 50% depending on assumptions about particle density, which are affected by the $m$-$D$ relation. Later studies (e.g., Mitchell, 1996; Erfani and Mitchell, 2016) attributed differences in model output to the influence of particle mass on terminal fall velocities.

Although many studies have established *m-D* relations for specific cases, a universal *m-D* relationship has not been found nor can a single relation be expected to represent the wide range of crystal habits and sizes within clouds occurring at different temperatures, locations, or formed by different mechanisms. Moreover, a single relationship cannot account for the natural variability of cloud properties such as particle size, shape, and density that occurs even in similar environmental conditions. Thus, some alternate approach is more appropriate for modeling and remote sensing studies that considers multiple *m-D* relations over many retrievals or model simulations to evaluate the variability in the ensemble results.

While previous studies (e.g., McFarquhar et al., 2007b; Heymsfield et al., 2010; Mascio et al., 2017) have considered how *m-D* relations vary with environmental conditions, such as temperature, the derived relations were fixed regardless of potential fluctuations for that environment. Further uncertainties were associated with measurement errors induced by shattering of large ice crystals on probe tips and subsequent detection within the probe's sample volume (Field et al., 2003), the processing techniques used (McFarquhar et al., 2017), and from the statistical counting of particles (e.g., Hallett, 2003; McFarquhar et al., 2007a). The approach by Fontaine et al. (2014) evaluated the variability in the prefactor *a* for an assumed exponent *b* for two field projects, but ultimately still derived a single *m-D* relationship for each dataset based on the mean conditions.

Extending the approach of McFarquhar et al. (2015), which derived a volume of equally realizable solutions within the phase space of the three gamma fit parameters (concentration $N_0$, shape $\mu$, and slope $\lambda$) characterizing PSDs, a novel approach is used here to determine equally valid *m-D* relations for a given environment. Data from a variety of environments sampled during the Mid-latitude Continental Convective Clouds Experiment (MC3E) are used to establish a surface of equally plausible *a* and *b* coefficients in (*a,b*) phase space using a technique that minimizes the chi-square difference between the *TWC* and *Z* derived from the PSDs measured by optical array probes (OAPs) and that directly measured by a *TWC* probe and radar.

The remainder of this paper is organized as follows. Section 2 outlines the datasets used and the methodology to process the radar and microphysics data, while Sect. 3 describes the technique employed to determine the surfaces of *m-D* coefficients. A brief description of the MC3E cases used in this study is provided in Sect. 4, and the surfaces of coefficients are derived and discussed in Sect. 5. A summary of the technique and its implications for numerical modeling and remote sensing retrieval schemes are given in Sect. 6.

## 2   Data and methodology

The data in this study were collected within mesoscale convective systems (MCSs) during the 2011 Mid-latitude Continental Convective Clouds Experiment (MC3E; Jensen et al., 2016). The study presented here uses data from cloud microphysical instruments aboard the University of North Dakota (UND) Cessna Citation II aircraft and from the Vance Air Force Base, OK (KVNX) Weather Surveillance Radar-1988 Doppler (WSR-88D) radar.

### 2.1   Identification of coincident aircraft/radar data

The use of airborne microphysical measurements and radar data collected from the ground allowed sampling of the same region of the cloud from microphysical and remote sensing perspectives. Use of the Airborne Weather Observation Toolkit (https:/

/github.com/swnesbitt/AWOT) radar matching algorithm and the Python ARM Radar Toolkit (Py-ART; Helmus and Collis, 2016) permitted calculation of radar $Z$ in the vicinity of the aircraft for each second of in situ cloud distributions measured during flight. The algorithm organizes all radar gates in a 3-Dimensional space (Maneewongvatana and Mount, 1999) for efficient acquisition of radar parameters at nearby radar range gates. The Barnes (1964) interpolation technique is then applied to data at the eight nearest gates within 500 m of the aircraft's location, ignoring vertically adjacent gates beyond a range of 65 km as the beamwidth exceeds the 500 m threshold, to obtain an averaged $Z$ at the aircraft location.

To compare microphysical properties with radar-measured Z for constant altitude flight legs at similar environmental temperature, only those times when the radar and microphysical datasets are coincident and the temperature varies by less than 1 °C were considered. To reduce uncertainty due to counting statistics in the measured PSDs, microphysical data were averaged over a 10 s period. Each 10 s period determined required radar echo and microphysical data for all 1 s samples to ensure that the aircraft and matched radar $Z$ were completely in cloud during the 10 s period. The *TWC* measurements and matched radar $Z$ were then averaged over the same 10 s period, with each 10 s interval assigned as a coincident point. Table 1 lists the start and end times, mean altitude, and temperature for each of the 16 constant-temperature flight legs flown when the UND Citation was in cloud. Observations where the mean *TWC* for a 10 s interval $< 0.05$ g m$^{-3}$ were ignored as the values were considered either below the noise threshold of the Nevzorov probe or optically thin cloud. To further constrain the study to periods when clouds were dominated by ice phase hydrometeors such that *TWC* $\approx$ *IWC* and to reduce the impact of liquid phase hydrometeors on the derived *TWC* and *Z*, observations were excluded from the analysis if the concentration from the cloud droplet probe exceeded 10 cm$^{-3}$ at any point during the 10 s interval which usually corresponds to the presence of water (Heymsfield et al., 2011). Of the coincident observations considered, 13% were excluded from the analysis based on these criteria. A total of 489 coincident observations were retained for this analysis.

## 2.2 Radar measurements

Data from the KVNX S-band (10 cm wavelength) radar were used in this study. Although the NASA dual-polarization (N-Pol) S-band Doppler radar was deployed during MC3E, mechanical issues prevented reliable collection of data for two of the three events examined here. Radars at other wavelengths collected data during MC3E. However, attenuation through liquid portions of the cloud (e.g., Bringi et al., 1990; Park et al., 2005; Matrosov, 2008) and non-Rayleigh scattering by larger particles (e.g., Lemke and Quante, 1999; Matrosov, 2007) could not be accounted for, and prompted exclusive use of the S-band radar.

Radar reflectivity factor values for gates near the UND Citation (Sect. 2.1) were used to obtain the average value of $Z$ using the radar matching algorithm only if the following criteria were met: correlation coefficient $\rho_{HV} \geq 0.75$, sigma differential phase $SDP \leq 12$ deg$^2$ km$^{-2}$, differential reflectivity $-2 \leq Z_{DR} \leq 3$ dB, and reflectivity texture (defined as the standard deviation in $Z$ of the nearest 5 gates) $< 7$ dBZ. These ranges represent acceptable values for echoes based on previous studies (Bringi and Chandrasekar, 2001). Radar gates not meeting these criteria were masked, reducing the likelihood of including gates with excessive signal noise due to clutter or weak signal, contamination by the aircraft, or other factors. For instances where the matched $Z$ changed by more than 2 dBZ for subsequent 1 s points (fewer than one percent of the observations), all radar gates factored into the radar matching algorithm were inspected by eye to ensure that no outlier values were responsible

for the jump in the matched *Z*. Of the observations that were manually inspected, all appeared spatially consistent with no outliers present, and as such remained in the averaging routine of the matching algorithm discussed in Sect. 2.1.

## 2.3 Microphysical measurements

During MC3E the Citation aircraft sampled clouds in situ, with most data collected in ice phase clouds between the melting layer and cloud top (Jensen et al., 2016). A suite of microphysical instruments was installed on the aircraft, including OAPs, which were used to image particles and derive PSDs, and a TWC probe. Specifics on the instrumentation and steps used to process the data are described below.

### 2.3.1 OAP data

A cloud imaging probe (CIP), a 2D cloud (2D-C) probe, and a High Volume Precipitation Spectrometer Version 3 (HVPS-3) sized particles by shadowing photodiode arrays attached to fast response electronics. Data from the 2D-C and HVPS-3 were combined to create a composite PSD, permitting particles between 150 µm and 19.2 mm to be considered in the analysis. The 2D-C was used instead of the CIP in the analysis even though the CIP has a larger sample volume because the inclusion of anti-shattering tips on the 2D-C reduced the impact of shattered artifacts (e.g., Korolev et al., 2011). Previous studies (Korolev et al., 2011, 2013a; Jackson et al., 2014) have shown that use of algorithms to identify shattered artifacts are sometimes needed even when the OAP is equipped with anti-shattering tips. Artifacts are identified by examining the frequency distribution of the times between which particles enter the sample volume (inter-arrival time; Field et al., 2006). When artifacts are present, this distribution follows a bimodal distribution with naturally-occurring particles having a mode with longer inter-arrival times and shattered artifacts having a mode with shorter inter-arrival times (e.g., Field et al., 2003). During MC3E there was only one mode in the inter-arrival time distribution corresponding to the naturally-occurring particles (Wu and McFarquhar, 2016) at all times, suggesting there were few shattered artifacts. Therefore, no shattering removal algorithm was used for the 2D-C and HVPS. Following Wu and McFarquhar (2016), the number distribution function *N(D)* was determined using the 2D-C for particles with $D < 1$ mm and the HVPS-3 for $D > 1$ mm. The 1 mm cutoff was chosen since *N(D)* for the two OAPs agreed on average within 5 percent for $0.8 \leq D \leq 1.2$ mm, and was used for all PSDs irrespective of periods when the difference between *N(D)* for the OAPs exceeded 5% in the overlap region. Given uncertainties in the probe's sample area and limitations of its depth of field for smaller particle sizes (Baumgardner and Korolev, 1997), particles with $D < 150$ µm were not included in the analysis.

The OAP data were processed using the University of Illinois/Oklahoma OAP Processing Software (UIOOPS; McFarquhar et al., 2018). Numerous morphological properties were calculated (e.g., particle maximum dimension, projected area, perimeter, area ratio, and habit) for individual particles, and PSDs were determined for each second of flight. Following Heymsfield and Baumgardner (1985) and Field (1999), only particles imaged with their center within the OAP's field of view were considered as otherwise there is too much uncertainty in particle size. Particles were identified as having their center within the field of view if their maximum dimension along the time direction exceeded the largest length where the particle potentially touched the edge of the photodiode array.

### 2.3.2 TWC data

The *TWC* was determined from the Nevzorov probe using the power required to melt or evaporate ice particles impinging on the inside of a cone (e.g., Nevzorov, 1980; Korolev et al., 1998). The probe used had a deeper cone than previous designs with a $60°$ vertex angle (as opposed to a $120°$ angle) that prevented many particles from bouncing out of the cone. Because previous
studies suggested that particles with $D > 4$ mm can bounce out of even the deeper cone (Wang et al., 2015), *TWC* may be underestimated when such particles are present. However, Korolev et al. (2013b) showed that the ratio of the Nevzorov *IWC* to that derived from the measured PSDs using the BF95 relation did not significantly vary with particle maximum dimension. Of the coincident points belonging to constant altitude flight legs in this study, $79.2\%$ of the observations had cumulative mass estimates using the BF95 relation from particles with $D \leq 4$ mm contributing at least $80\%$ to the total mass. Therefore,
measurements of *TWC* were included irrespective of whether $D_{\max} > 4$ mm.

## 3   Development of equally plausible (*a,b*) surfaces

In this section, a method for determining a surface of equally realizable solutions for *m-D* coefficients in the phase space of (*a,b*) coefficients is described. The surface of these coefficients is determined through a procedure that minimizes the $\chi^2$ differences between the *TWC* and *Z* derived from *N(D)* and that directly measured by the Nevzorov and ground-based radar, respectively.
The minimization procedure is carried out for each constant-temperature flight leg (defined by temperature varying by less than $1 °C$) for the MC3E cases studied. This approach follows that of McFarquhar et al. (2015) who developed volumes of equally realizable $N_0$, $\mu$, and $\lambda$ characterizing observed *N(D)* as gamma distributions for observations obtained during the Indirect and Semi-Direct Aerosol Campaign (ISDAC) and the NASA African Monsoon Multidisciplinary Analyses project (NAMMA).

For an individual 10 s sample, the *TWC* and *Z* derived from the PSD for a specific *a* and *b* is given by $TWC_{\mathrm{SD}}$ and $Z_{\mathrm{SD}}$,
respectively, as

$$TWC_{\mathrm{SD}} = \sum_{j=1}^{N}(aD^b)N(D_j)dD_j \text{ and} \tag{1}$$

$$Z_{\mathrm{SD}} = \Big(\frac{6}{\pi\rho_{ice}}\Big)\frac{|K_{ice}|^2}{|K_w|^2}\sum_{j=1}^{N}(aD^b)^2N(D_j)dD_j \tag{2}$$

following the method of Hogan et al. (2006) and accounting for the different dielectric constants for water ($|K_w|^2 = 0.93$)
and ice ($|K_{ice}|^2 = 0.17$). Uncertainties in $TWC_{\mathrm{SD}}$ and $Z_{\mathrm{SD}}$ are discussed later in this section. The metric defining the difference between the *TWC* and *Z* derived from *N(D)* for a specific *a* and *b* and that directly measured by the Nevzorov and ground-based radar, respectively, is given by $TWC_{\mathrm{diff}}$ and $Z_{\mathrm{diff}}$ as follows:

$$TWC_{\mathrm{diff}} = \left[\frac{TWC - TWC_{\mathrm{SD}}(a,b)}{\sqrt{TWC \times TWC_{\mathrm{SD}}(a,b)}}\right]^2 \text{ and} \tag{3}$$

$$Z_{\text{diff}} = \left[ \frac{\sqrt{Z} - \sqrt{Z_{\text{SD}}(a,b)}}{\sqrt{\sqrt{Z} \times \sqrt{Z_{\text{SD}}(a,b)}}} \right]^2 . \tag{4}$$

In this study, $TWC_{\text{diff}}$ and $Z_{\text{diff}}$ are computed for all points in the domain of values encompassing $5 \times 10^{-4} < a < 0.35$ g cm$^{-b}$ and $0.20 < b < 5.00$ at increments of $5 \times 10^{-4}$ g cm$^{-b}$ and 0.01, respectively.

5  Given a priori assumptions of $Z$ being proportional to the square of a particle's mass, the square root of reflectivity was used in Eq. (4) so that $TWC_{\text{diff}}$ would be similar to $Z_{\text{diff}}$ on average and each would have approximately equal weight in determining $a$ and $b$. Although radar $Z$ measurements involve a significantly greater sample volume than that of OAPs and a bulk content probe, $TWC_{\text{diff}}$ and $Z_{\text{diff}}$ were not weighted proportionally to the sample volume in order to ensure that both bulk moments had some impact on the derived $a$ and $b$. Given that larger ice crystals are fractionally more important than small crystals

10 in determining $Z_{\text{SD}}$ than $TWC_{\text{SD}}$ and given varying contributions of larger crystals to $Z_{\text{SD}}$ and $TWC_{\text{SD}}$, $TWC_{\text{diff}}$ has a greater impact on the $\chi^2$ minimization procedure some of the time while $Z_{\text{diff}}$ does at other times. The ratios between $Z_{\text{diff}}$ and $TWC_{\text{diff}}$ for each flight leg are given in Table 2, and range between 0.32 and 8.58 with a mean of 2.62 for the 16 flight legs. No attempt is made to force equal weight for $Z_{\text{diff}}$ and $TWC_{\text{diff}}$ for each coincident point because there are periods when cloud properties influence $TWC$ differently than $Z$.

15  At first, the sum of $TWC_{\text{diff}} + Z_{\text{diff}}$ is used to identify $(a,b)$ values that characterize an individual 10 s data point. An example of $TWC_{\text{diff}} + Z_{\text{diff}}$ computed in $(a,b)$ phase space for a 10 s averaged PSD measured beginning at 13:56:45 UTC on 20 May 2011 is shown in Fig. 2a. The color representing $TWC_{\text{diff}} + Z_{\text{diff}}$ is shaded on a logarithmic scale to more easily show the range of values. The smallest swath of values, arbitrarily chosen as being $TWC_{\text{diff}} + Z_{\text{diff}} \leq 1$ within the region outlined black, spans $b$ values of 1.13 to 4.72. The curvature in the outlined region highlights the correlation of $a$ and $b$ showing that similar

20 $m$ can be obtained using very different $b$ by adjusting $a$ accordingly. Considering both $TWC_{\text{diff}}$ and $Z_{\text{diff}}$ allows the shape and placement of the smallest swath of values to adjust according to two different moments of the PSD since conditions impact $TWC$ differently than $Z$. Using two constraints on the $\chi^2$ minimization technique therefore provides additional insight into the microphysical properties as discussed in Sect. 5.

  The chi-square statistic for a flight leg, defined as

25 $$\chi^2(a,b) = \frac{1}{N} \sum_{i=1}^{N} \left[ TWC_{\text{diff}}(i) + Z_{\text{diff}}(i) \right], \tag{5}$$

involves a summation over all N 10 s coincident observations represented by the index $i$ and normalized by N. When $\chi^2$ is computed by summing over all N points in the flight leg, the region with the smallest $\chi^2$ ($\chi^2 \leq 1$; outlined region in Fig. 2b) is smaller than the region in Fig. 2a which shows $\chi^2$ for a single point, because different $(a,b)$ minimize $\chi^2$ for each of the individual PSDs in the 5 minute period depicted. Therefore, overall the $\chi^2$ values are higher than the $TWC_{\text{diff}} + Z_{\text{diff}}$ computed

30 for each $(a,b)$. The point in Fig. 2b corresponds to the $a$ and $b$ point that minimizes $\chi^2$, hereafter represented as $\chi^2_{\text{min}}$, which represents the most likely $a$ and $b$ value.

To represent the uncertainty in the derived coefficients for each flight leg, all $a$ and $b$ fulfilling $\chi^2 \leq \chi^2_{\min} + \Delta\chi^2$ are assumed to be equally plausible solutions. Analogous to McFarquhar et al. (2015), the confidence region is defined as $\Delta\chi^2 = \max(\chi^2_{\min}, \Delta\chi^2_1, \Delta\chi^2_2)$. The $\chi^2_{\min}$ characterizes the robustness of the minimization procedure affected by the natural parameter variability over a flight leg, $\Delta\chi^2_1$ represents uncertainties in the PSD due to statistical sampling uncertainties, and $\Delta\chi^2_2$ represents measurement uncertainties. Similar to their study, $\Delta\chi^2_1$ is determined here as

$$\Delta\chi^2_1 = \frac{1}{N}\sum_{i=1}^{N} \frac{1}{2}\left\{\left[\frac{TWC_{SD,\min}(i)-TWC_{SD}(i)}{\sqrt{TWC_{SD,\min}(i)\times TWC_{SD}(i)}}\right]^2 + \left[\frac{\sqrt{Z_{SD,\min}(i)}-\sqrt{Z_{SD}(i)}}{\sqrt{\sqrt{Z_{SD,\min}(i)}\times\sqrt{Z_{SD}(i)}}}\right]^2\right\} + \frac{1}{2}\left\{\left[\frac{TWC_{SD,\max}(i)-TWC_{SD}(i)}{\sqrt{TWC_{SD,\max}(i)\times TWC_{SD}(i)}}\right]^2 + \left[\frac{\sqrt{Z_{SD,\max}(i)}-\sqrt{Z_{SD}(i)}}{\sqrt{\sqrt{Z_{SD,\max}(i)}\times\sqrt{Z_{SD}(i)}}}\right]^2\right\}. \tag{6}$$

The different terms in Eq. (6) represent the difference in the minimum and maximum $TWC$ or $Z$ derived from the minimum and maximum $N(D)$ using the most likely $(a,b)$ minimizing $\chi^2$ ($TWC_{SD,\min}$ and $TWC_{SD,\max}$ or $Z_{SD,\min}$ and $Z_{SD,\max}$) and that derived from the measured $N(D)$ ($TWC_{SD}$ or $Z_{SD}$). Following McFarquhar et al. (2015), the minimum and maximum $N(D)$ are determined by subtracting or adding the square root of the number of particles counted in each size bin to the number of particles counted in the bin when computing $N(D)$. This technique represents uncertainty in the actual particle counts for each size bin as given by Poisson statistics (Hallett, 2003; McFarquhar et al., 2007a).

Estimates of the measurement uncertainty from the OAPs, Nevzorov probe, and ground-based radar also influence the uncertainty in the derived coefficients. The uncertainty due to measurement error $\Delta\chi^2_2$ is defined as

$$\Delta\chi^2_2 = \frac{1}{N}\sum_{i=1}^{N} \frac{1}{2}\left\{\left[\frac{TWC_{SD,\text{meas\_min}}(i)-TWC_{SD}(i)}{\sqrt{TWC_{SD,\text{meas\_min}}(i)\times TWC_{SD}(i)}}\right]^2 + \left[\frac{\sqrt{Z_{SD,\text{meas\_min}}(i)}-\sqrt{Z_{SD}(i)}}{\sqrt{\sqrt{Z_{SD,\text{meas\_min}}(i)}\times\sqrt{Z_{SD}(i)}}}\right]^2 + \left[\frac{TWC_{\text{meas\_min}}(i)-TWC(i)}{\sqrt{TWC_{\text{meas\_min}}(i)\times TWC(i)}}\right]^2 + \left[\frac{\sqrt{Z_{\text{meas\_min}}(i)}-\sqrt{Z(i)}}{\sqrt{\sqrt{Z_{\text{meas\_min}}(i)}\times\sqrt{Z(i)}}}\right]^2\right\} + \frac{1}{2}\left\{\left[\frac{TWC_{SD,\text{meas\_max}}(i)-TWC_{SD}(i)}{\sqrt{TWC_{SD,\text{meas\_max}}(i)\times TWC_{SD}(i)}}\right]^2 + \left[\frac{\sqrt{Z_{SD,\text{meas\_max}}(i)}-\sqrt{Z_{SD}(i)}}{\sqrt{\sqrt{Z_{SD,\text{meas\_max}}(i)}\times\sqrt{Z_{SD}(i)}}}\right]^2 + \left[\frac{TWC_{\text{meas\_max}}(i)-TWC(i)}{\sqrt{TWC_{\text{meas\_max}}(i)\times TWC(i)}}\right]^2 + \left[\frac{\sqrt{Z_{\text{meas\_max}}(i)}-\sqrt{Z(i)}}{\sqrt{\sqrt{Z_{\text{meas\_max}}(i)}\times\sqrt{Z(i)}}}\right]^2\right\}. \tag{7}$$

The terms $TWC_{SD,\text{meas\_min}}$, $TWC_{SD,\text{meas\_max}}$, $Z_{SD,\text{meas\_min}}$, and $Z_{SD,\text{meas\_max}}$ represent the minimum and maximum $TWC$ or $Z$ derived using a 50% uncertainty in the measured $N(D)$. This uncertainty follows Heymsfield et al. (2013) where up to a 50% difference in the number concentration for particles with $D > 0.1\,\mathrm{mm}$ was determined. Uncertainties in the bulk measurements of $TWC$ and $Z$ must also be considered in the generation of the uncertainty surfaces with the minimum and maximum possible bulk values represented as $TWC_{\text{meas\_min}}$, $TWC_{\text{meas\_max}}$, $Z_{\text{meas\_min}}$, and $Z_{\text{meas\_max}}$. Following Korolev et al. (2013b), it was assumed that there was a 2% uncertainty when $D_{\max} \leq 4\,\mathrm{mm}$ and a 8% uncertainty for other periods to address the possibility of particles bouncing out of the cone of the Nevzorov probe. A radar reflectivity uncertainty of 1 dB (Krajewski and Ciach, 2003) is subtracted from or added to the measured $Z$ to determine $Z_{\text{meas\_min}}$ and $Z_{\text{meas\_max}}$.

Figure 3 illustrates the frequency distribution of the ratio between $\chi^2_{\min}$ and $\Delta\chi^2_1$ (blue shading) and between $\chi^2_{\min}$ and $\Delta\chi^2_2$ (red shading) for all 16 flight legs. Of all 16 legs considered, 15 have a ratio between $\chi^2_{\min}$ and $\Delta\chi^2_1$ greater than 1, meaning $\chi^2_{\min} > \Delta\chi^2_1$, and 50% of the observations have ratios greater than 10. For 5 of the 16 legs, the ratio between $\chi^2_{\min}$

and $\Delta\chi_2^2$ is greater than 1 indicating that the $\chi^2$ obtained from the ($a$,$b$) minimization procedure is greater than the difference between moments derived from the minimum and maximum $N(D)$ and from the minimum and maximum $TWC$ and $Z$ due to measurement errors for nearly a third of the periods in this study. This means that the natural parameter variability over a flight leg is sometimes more important for the derived uncertainty of $m$-$D$ coefficients, whereas at other times measurement errors are more important. This is further discussed in Sect. 5.

At first, the $b$ coefficients greater than 3 shown in Fig. 2 may seem counter intuitive as the mass of a particle cannot be greater than that of an ice sphere. Further, a particle's density would increase with increasing $D$ for $b > 3$. But, due to the covariability of $a$ and $b$, $b > 3$ does not necessarily imply the particle has a mass greater than a sphere. Nevertheless, equally plausible $b$ values greater than 3 were closely inspected as past studies (e.g., Fontaine et al., 2014) have disregarded $b > 3$ as a possible exponent in an $m$-$D$ relation. To investigate the impact of $b > 3$, a linear sequence of $b$ values in the plausible surface was generated for each flight leg and the 5[th], 25[th], 50[th], 75[th], and 95[th] percentiles of $b$ were determined. The corresponding $a$ from each of these $b$ was identified, and the cumulative reflectivity distribution functions, defined as

$$Z_c(D) = \left(\frac{6}{\pi \times \rho_{\text{ice}}}\right)^2 \frac{|K_{\text{ice}}|^2}{|K_w|^2} \int_0^D (aD'^b)^2 \, N(D') \, dD', \tag{8}$$

were computed using the mean $N(D)$ for the period and the particle mass derived with these $a$ and $b$. Figure 4 shows an example of the $Z_c(D)$ over the range of particle sizes observed from the -23 °C flight leg on 20 May 2011 using these $a$ and $b$ coefficients. The $Z_c(D)$ derived using BF95 coefficients, with the prefactor $a$ ($= 0.002 \text{ g cm}^{-1.9}$) modified following the correction factor of Hogan et al. (2012) applicable for the definition of $D$ used here, is also shown for reference. It is worth noting that the modified BF95 coefficients may reasonably resolve the particle mass for *some* particle sizes for the PSD depicted in Fig. 4. While the lower values of $a$ and $b$ yield larger $Z_c(D)$ for smaller $D$ than do the larger values of $a$ and $b$, the derived total reflectivity $Z_t = \int_{D_{min}}^{D_{max}} Z(D) \, dD$ for the 5[th] and 95[th] percentiles of $b$ are within $11.38 \text{ mm}^6 \text{ m}^{-3}$ of the mean matched radar $Z$ of 18.36 $\text{mm}^6 \text{ m}^{-3}$ (12.64 dBZ), a difference of 62 percent of the mean. In contrast, the difference of the mean from the $Z_t$ computed with modified BF95 coefficients is much higher, 88.6%, suggesting values of $b > 3$ are indeed giving plausible results for the range of particle sizes observed.

When the seven flight legs that have some values of $b > 3$ in the surface of equally plausible solutions are considered, $Z$ values for the 5[th] and 95[th] percentiles of $b$ are within 82.4% of the mean matched radar $Z$. While this value is greater than the 50.5% difference for the other flight legs and for the period illustrated in Fig. 4, $Z$ values for the 5[th] and 95[th] percentiles are more consistent with the mean matched radar $Z$ compared to that computed with the modified BF95 relationship.

Thus, the bulk variables such as $Z$ derived using $b > 3$ are physically plausible for the distributions examined here given the covariability of $a$ and $b$. However, this conclusion may only apply when the coefficients are applied over the range of particle sizes observed during MC3E and assuming PSDs with similar shapes. For example, for the 95[th] percentile of $b$ ($b = 3.61$) and the corresponding value of $a$ used to construct Fig. 4, ice particles with $D < 3.83$ cm have particle masses less than those of spherical particles with a density of solid ice for the same maximum dimension. On the other hand, if the covariability of $a$ and $b$ was not taken into account when choosing the corresponding $a$ value, then a particle could have a mass greater than that of a

spherical particle for much smaller $D$. While the technique highlights the possibility of a wide range of $m$-$D$ coefficients for a given environment, equally plausible solutions containing $b > 3$ are still not considered in the remainder of this study to remain consistent with previous studies and to avoid any chance of unphysical behavior should the equally plausible coefficients be extrapolated to PSDs from remote sensing retrievals or microphysics parameterization schemes that extend to particle sizes
larger than in the original dataset.

## 4   Events overview

The Citation aircraft sampled different ice phase environments during the 25 April, 20 May, and 23 May 2011 flights. Jensen et al. (2016) provide an overview of all MC3E cases, while Jensen et al. (2014) give a synoptic scale overview of the MCSs examined in this study. These particular events were chosen because of variations in how the complex of storms evolved and
the location of in situ measurements relative to the convective system. Figure 5 shows a 0.5° plan-position indicator (PPI) scan of corrected radar reflectivity from the KVNX radar for each event. The PPI was obtained during the middle of the UND Citation flight leg depicted by the black line in Fig. 5.

The first event involved an upper-level trough that produced ascent aloft and generated thunderstorms across northern Oklahoma around 06 UTC on 25 April 2011. As these storms traversed northward along an elevated frontal boundary overnight,
their bases decoupled from the boundary layer as daytime solar radiation ceased. The discrete cells evolved into an MCS and moved into southern Kansas by 11 UTC (Fig. 5a) when the Citation sampled weaker embedded convection and broader stratiform precipitation. The second MCS, with a north-to-south oriented squall line which was part of a larger system, developed from a line of convective cells originating in west Texas along a dry line around 10 UTC on 20 May 2011 and propagated into the deployment region in north central Oklahoma. The Citation aircraft primarily flew within the trailing stratiform region of
the MCS (Fig. 5b). The third MCS originated as a series of discrete supercell thunderstorms along a surface dry line in western Oklahoma and moved eastward into the MC3E domain by 21 UTC on 23 May 2011 before transitioning to a more linear MCS feature. Microphysical measurements were made in the anvil region of these strong thunderstorms (Fig. 5c).

To provide context of the bulk characteristics sampled during each event, boxplots of $Z$ matched at the aircraft's location and $TWC$ from the Nevzorov probe for each constant-temperature flight leg are given in Fig. 6. The whiskers represent the
$5^{th}$ and $95^{th}$ percentiles from coincident observations, the box edges denote the $25^{th}$ and $75^{th}$ percentiles, and the red line in the middle is the median. Distributions are listed in order of decreasing temperature, with instances of multiple legs having the same average temperature shown in chronological order. While the bulk $TWC$ and $Z$ may differ for flight legs of similar average temperature on a given day, as in the -26.5 and -35 °C environments on 25 April (Figs. 6a-b), greater or smaller $TWC$ correlates with greater or smaller $Z$ for most cases. The variability in the $TWC$ and $Z$ as it relates to the construction of surfaces
of equally plausible $m$-$D$ coefficients is discussed in the next section.

## 5 Results

This section discusses how the (*a,b*) surfaces vary between different cases, as a function of temperature, depending on the determination of radar reflectivity, and depending on whether PSDs had large mass contributions from particles with $D > 4$ mm.

### 5.1 Radar absolute *Z* calibration

While S-band radars within the NEXRAD WSR-88D network are calibrated individually and among one another upon initial installation, biases in *Z* can develop over time (Ice et al., 2017). Zhang et al. (2013) described a technique that uses self-similarity in the *Z*, $Z_{DR}$, and specific differential phase ($K_{DP}$) fields to estimate the absolute *Z* bias for events in rain. This method was employed for the cases in this study and biases in *Z* of -1.08 (25 April), -0.65 (20 May), and 1.43 dBZ (23 May 2011) were found. These corrections were applied to the value of *Z* calculated as explained in Sect. 3. The surfaces of *m-D* coefficients derived using the matched radar *Z* and that with the bias corrections applied were similar, with the range of equally plausible *b* values differing, on average, by 6.4% after the corrections were made.

### 5.2 Accounting for mass contributions from larger particles

As discussed in Sect. 2.3.2, the Nevzorov probe is prone to larger particles ($D > 4$ mm) bouncing out of the collection cone resulting in potential *TWC* underestimations. Mass contents were derived from the PSDs using the modified BF95 coefficients to identify time periods in which the contribution of mass from particles with $D > 4$ mm was likely greater than 20%. Of all 10 s PSDs used in this study, 20.9% had mass contributions from these larger particles exceeding 20% of the total mass. Figure 7 illustrates the similarity in the (*a,b*) surfaces generated using all coincident observations (red shading) and only those using observations with mass from larger particles contributing $\leq 20\%$ of the total mass (blue shading) for the 23 May 2011 event. Regions of overlap between the two approaches only appear as purple shading. The sensitivity test shows that omitting observations where larger particles contribute fractionally more to the total mass yield an area of equally plausible (*a,b*) surfaces for the 23 May event differing, on average, by 1.4%. As such, all coincident observations are used for this study irrespective of the fractional contributions of particles with $D > 4$ mm to the mass.

### 5.3 Environmental impact on *m-D* coefficients

Surfaces of equally plausible *m-D* coefficients in (*a,b*) phase space from all flight legs outlined in Table 1 are shown in Fig. 8. For each event, flight legs are grouped by the same environmental temperature with the different colors corresponding to the time periods given in each panel. These surfaces are influenced by how *TWC* and *Z* derived from the PSDs relate to observed *TWC* and *Z*, and by the variability of each within a flight leg. The observed trends in the (*a,b*) surfaces and how they are affected by *N(D)*, *TWC*, and *Z* are discussed further below.

To compare surfaces of equally plausible solutions between different environments and also between periods with the same temperature, the percent of overlap between any two flight legs is computed and shown as a matrix in Fig. 9. The percentage

of overlap is determined by counting the number of ($a$,$b$) pairs contained in both equally plausible surfaces for the conditions listed in the row and column in the matrix and dividing by the number of ($a$,$b$) pairs in the surface for the condition listed in the row multiplied by 100%. There are two values in the matrix corresponding to each comparison between two flight legs, with differences between the two values resulting from dividing the area of the equally plausible surface from the corresponding

column by that in the corresponding row in the matrix. Thus, it is possible for the percentage of overlap between two flight legs to be greater when normalized by an equally plausible surface that is smaller in area, and to be smaller when normalized by a larger equally plausible surface. It is worth noting that the percentage of overlap does not always follow an organized trend with respect to moving away from the gray diagonal line in the matrix as depicted in the top right corner of Fig. 9a. The lack of organized overlap values in some regions of the matrix could be influenced by the sensitivity in computing the overlap region

over a fine resolution of ($a$,$b$) values within the domain described in Sect. 3, or perhaps could change in a more organized manner if there was a more statistically representative sample for these calculations to be made. Using the ($a$,$b$) surfaces from the -26.5 °C flight legs on 25 April (Fig. 8b) as an example, 62% of the ($a$,$b$) surface for the 11:05:20–11:14:45 UTC period (labeled -26.5 °C I; Fig. 9a) overlaps with the later -26.5 °C flight leg while 65% of the ($a$,$b$) surface for the 11:21:20–11:34:05 UTC period (labeled -26.5 °C II) overlaps with the earlier -26.5 °C flight leg. The difference occurs because there are 1132

($a$,$b$) pairs in the surface for the 11:05:20–11:14:45 UTC period and 1077 ($a$,$b$) pairs in the surface for the 11:21:20–11:34:05 UTC period. Flight legs having the same temperature are ordered chronologically as in Fig. 8 and differentiated with a Roman numeral. Differences of the ($a$,$b$) surfaces between flight legs are further discussed below.

### 5.3.1   25 April case

While differences exist between the ($a$,$b$) surfaces for the near-constant temperature legs on 25 April (Fig. 9a), these surfaces

have considerable overlap with each other for $a < 0.01$ g cm$^{-b}$ and $b < 2.5$ (Figs. 8a-c). The -22 and -26.5 °C legs have similar sets of equally plausible solutions, with ($a$,$b$) surfaces overlapping between 46 and 91% (Fig. 9a). Less agreement in the ($a$,$b$) surfaces is observed among the -35 °C flight legs, with the surfaces overlapping on average 27.8% among the different combinations. The differences in the size of the surfaces is primarily influenced by the natural variability within cloud ($\Delta\chi^2 = \chi^2_{min}$) for 5 of the 7 legs and by the uncertainty due to measurement errors ($\Delta\chi^2 = \Delta\chi^2_2$) for the remaining legs. The areas

of the ($a$,$b$) surfaces for the -22 and -26.5 °C legs were, on average, 31.2% smaller than the surfaces associated with the -35 °C environment (Figs. 8a-c). Three of the four -35 °C legs have surfaces larger than the -22 and -26.5 °C environments as the surface of equally plausible $m$-$D$ coefficients extends beyond the maximum value $a$ of 0.017 g cm$^{-b}$ and $b$ of 3.00 found for the -22 and -26.5 °C legs. To explain the variation of these ($a$,$b$) surfaces for the different temperatures, the distributions of microphysical quantities for the times corresponding to these surfaces were examined.

To examine the variability in hydrometeors, particle images and distributions of bulk microphysical properties were analyzed for each flight leg. Example particle images from the HVPS-3, which provide information on the size and habit of ice phase particles with $D > 1$ mm, are plotted in Fig. 10. The pictured particles represent a subset of those imaged for the time period given and were chosen at random in an attempt to obtain a representative sample of hydrometeors. Figure 11 shows the mean $N(D)$ and cumulative mass distribution function $M(D)$ using the modified BF95 relationship for each flight leg analyzed in

this study. Figure 12 details the distribution of number concentration $N_t$, median mass diameter $D_{mm}$, and a metric for particle sphericity obtained from the PSDs derived from the 2D-C and HVPS-3 data at each 10 s coincident observation. The $D_{mm}$ is derived using the modified BF95 coefficients to compare among the different flight legs. The whiskers and box edges are the same as in Fig. 6. Particle sphericity $\zeta$ (McFarquhar et al., 2005; Finlon et al., 2016) is defined by

$$\zeta = A^{1/2}/P, \tag{9}$$

where $A$ is the cross-sectional area directly measured by the probe and $P$ is the perimeter determined from the sum of all pixels within one diode width of the edge of the particle and the diode resolution. Finlon et al. (2016) described how higher $\zeta$ denotes more-circular particles. Sphericity values shown in Fig. 12 represent a mass-weighted mean of $\zeta$ for all particles using mass estimated from the modified BF95 relation within each 10 s observation. Figures 10, 11, and 12 are ordered in the same manner
as in Fig. 6, with instances of multiple legs having the same average temperature shown in chronological order.

As evidenced by the particle images and mean $N(D)$ at T = -22 and -26.5 °C (Figs. 10a-c, 11a), the presence of aggregates exceeding 5 mm is more common compared to lower temperatures (Figs. 10d-g) where the ice crystals and aggregates appear to be skewed towards smaller sizes. Distributions of $D_{mm}$ (Fig. 12b) and $TWC$ (Fig. 6b) also indicate this trend, with a median $D_{mm}$ for the 11:05:20–11:14:45 UTC (T = -26.5 °C) flight leg of 2.2 mm while the -35 °C periods have median $D_{mm}$ ranging
between 1.1 and 1.7 mm.

To illustrate that the range of equally plausible (a,b) coefficients is sometimes explained more by the variability of cloud parameters than the uncertainty in measurement errors, the distributions of bulk microphysical variables, $TWC$, and $Z$ are compared between the 11:05:20–11:14:45 UTC (T = -26.5 °C) and 10:03:05–10:08:45 UTC (T = -35 °C) periods. The -26.5 °C flight leg had ranges in $N_t$, $D_{mm}$, sphericity, $Z$, and $TWC$ between the 25th and 75th percentiles (interquartile range hereafter)
of 21.5 L$^{-1}$, 1.3 mm, 0.04, 5.2 dBZ, and 0.73 g m$^{-3}$, respectively, while the same variables for the -35 °C period had smaller interquartile ranges of 7.4 L$^{-1}$, 0.1 mm, 0.02, 4.0 dBZ, and 0.17 g m$^{-3}$ (Figs. 6a,b; 12a-c). The distribution of $\chi^2$ in (a,b) phase space is expected to differ when the variability in $N(D)$ throughout a flight leg is different between two periods since different $a$ and $b$ are likely to yield $TWC_{SD}$ and $Z_{SD}$ similar to the observed $TWC$ and $Z$. Figure 13 illustrates the distribution of $\chi^2$ for the two periods, with the outlined region representing $\chi^2$ values that are $\leq 2$ for comparison. The region containing $\chi^2 \leq$
2 is 90.8% smaller for the -26.5 °C flight leg compared to the -35 °C period and indicates that the $TWC_{SD}$ and $Z_{SD}$ derived from all possible $a$ and $b$ remain fairly consistent over the course of the -26.5 °C flight leg due to the smaller interquartile ranges in the $TWC$, $Z$, and bulk microphysical properties. As such, low $\chi^2$ values are present over a larger range of $m$-$D$ coefficients for the -35 °C leg.

Although the distribution of $\chi^2$ is an important factor in determining the area of an equally plausible surface, the $\Delta\chi^2$
confidence region, which is equal to $\chi^2_{min}$ ($\Delta\chi^2_2$) for 4 (3) of the flight legs on this day, can also influence the area of (a,b) surfaces. While the allowable tolerance is a factor of 2 greater for the -26.5 °C leg, the equally plausible (a,b) surface is 3.4 times smaller compared to the -35 °C flight leg (Figs. 8b,c) because of the magnitude and distribution of $\chi^2$ values in (a,b) phase space. Put another way, more $\chi^2$ values considered within the (a,b) phase space are greater than the $\chi^2_{min} + \Delta\chi^2$ criteria to be considered equally plausible solutions compared to the -35 °C leg.

### 5.3.2    20 May case

The wide range of temperatures sampled during the 20 May event was associated with a large variation in $Z$ (Fig. 6c), with
median values ranging between 12.5 dBZ (T = -23 °C) and 27.1 dBZ (T = -5.5 °C). Representative particle images (Fig. 14)
highlight differences in particle size and habit between the higher temperature flight legs (T = -5.5 and -10.5 °C) and the
lower temperature periods (T = -16 and -23 °C), with images and mean $N(D)$ (Fig. 11b) from the -5.5 and -10.5 °C legs
indicating a greater frequency of larger ice crystals and aggregates with $D \geq 2$ mm. A Mann-Whitney U test confirms that
$D_{mm}$ (Fig. 12e) and sphericity (Fig. 12f) between the higher and lower temperature environments are statistically different at
the 99% confidence level, with notably larger and less spherical particles observed during the -5.5 and -10.5 °C flight legs.
Further, median $Z$ for the -5.5 and -10.5 °C periods (22.3–27.1 dBZ) are up to 30.7 times greater than for the -16 and -23 °C
legs (12.2–12.5 dBZ) while the median $TWC$ are up to 1.9 times (0.3 g m$^{-3}$) greater for the -5.5 and -10.5 °C legs. Thus, the
difference in particle properties and bulk properties $TWC$ and $Z$ can be used to explain differences in $(a,b)$ coefficients observed
between the legs on this day.

Microphysical properties such as the effective density $\rho_e$ of ice hydrometeors can impact $TWC$ differently than they do $Z$.
The $\rho_e$, defined here as the ratio of $TWC$ derived assuming the modified BF95 relationship to the integrated volume of particles
enclosed by an oblate spheroid with an aspect ratio of 0.6 (e.g., Hogan et al., 2012), is estimated to evaluate its influence on
$TWC$ and $Z$. Median $\rho_e$ ranges between 0.05 and 0.08 g cm$^{-3}$ for the -5.5 and -10.5 °C periods and between 0.18 and 0.21
g cm$^{-3}$ for the -16 and -23 °C flight legs. These trends along with minimal riming evident from the 2D-C particle images
suggest that particles are on average less compact for the higher temperature legs. Further, the presence of larger aggregates as
suggested by greater values of $D_{mm}$ (Fig. 12e), lower sphericity (Fig. 12f) and $\rho_e$, and the representative particle images from
the HVPS-3 (Figs. 14a,b) are consistent with increasing $Z$ when observed by longer wavelength radars (e.g., Giangrande et al.,
2016).

Since differences in $\rho_e$ appear to affect the $TWC$ and $Z$ on 20 May, the variability in $N(D)$ is not the only factor influencing
the equally plausible $(a,b)$ surfaces depicted in Figs. 8d-g. Figure 9b illustrates that only the -16 and -23 °C legs have similar
$(a,b)$ surfaces, with 85% of the $(a,b)$ coefficients from the -16 °C leg overlapping with the -23 °C flight leg. Minimum values
of $b$ for the -5.5 and -10.5 °C flight legs, where less compact particles were observed, were 1.84 and 1.66, respectively, while
minimum $b$ for the -16 and -23 °C legs were 1.09 and 1.06 for similar $a$ (Figs. 8d-g). Looking at the $(a,b)$ surfaces another
way, values of $a$ for the -5.5 and -10.5 °C legs were as large as 0.031 g cm$^{-b}$ while $a$ exceeds 0.05 g cm$^{-b}$ for $b = 3$ during
the -16 and -23 °C flight legs. Although the $\Delta\chi^2$ confidence region is equal to $\Delta\chi_2^2$ for the 4 flight legs on this day and has
$\Delta\chi^2$ values that are within 1% of each other, the distribution of $\chi^2$ greatly influences the extent of these surfaces in $(a,b)$ phase
space with an area for the -5.5 and -10.5 °C flight legs that is on average 2.9 times smaller than the the -16 and -23 °C periods.
When considering the $m = aD^b$ relation whose size $D$ and exponent $b$ are held fixed, lower values of $a$ as observed during the
-5.5 and -10.5 °C legs suggest that particles on average have smaller $m$ compared to the -16 and -23 °C legs and are consistent
with smaller $\rho_e$ observed for the -5.5 and -10.5 °C periods.

### 5.3.3   23 May case

The 23 May case was unique from the other two cases in that the bulk $Z$ varied less between the different temperature environments (Fig. 6e), with median $Z$ ranging only between 16.9 and 18.2 dBZ. Representative particle images (Fig. 15) in addition to the mean $N(D)$ (Fig. 11c) and the cumulative $M(D)$ (Fig. 11f) indicate that the sizes and shapes of ice hydrometeors are

similar for all five flight legs. Additionally, distributions of $D_{mm}$ (Fig. 12h) and sphericity (Fig. 12i), with median values of each varying by 0.4 mm and 0.04 respectively, further support this similarity in cloud properties between the different environments. Equally plausible $(a,b)$ surfaces were also similar irrespective of temperature (Figs. 8h,i), with the four flight legs after the 21:49:55–21:55:15 UTC period having surfaces that overlap on average 62.1% among the different combinations (Fig. 9c). The 21:49:55–21:55:15 UTC leg is the only period on this day where the $\Delta\chi^2$ confidence region is determined by the natural

variability in the cloud ($\chi^2_{\min}$) rather than the uncertainty due to measurement errors ($\Delta\chi^2_2$). As such, the $(a,b)$ surface for this period has minimal overlap with the other equally plausible surfaces. Closer examination of the bulk $TWC$ (Fig. 6f) indicates that values at the fifth percentile for the 21:49:55–21:55:15 UTC period are 65.2% less than the remaining flight legs, which impacts the distribution of $\chi^2$ values and the $(a,b)$ values that are within the $\chi^2_{\min} + \Delta\chi^2$ threshold.

Although surfaces of equally plausible solutions trend larger in area for lower temperature environments on 25 April and 20

May, the area of $(a,b)$ surfaces among the five flight legs on 23 May are on average 2.2 (3.8) times smaller compared to the 25 April (20 May) event. To examine how the distribution of $\chi^2$ in $(a,b)$ phase space is affected by differences in the variability of $TWC$ and $Z$ throughout a flight leg, the 14:16:30–14:32:15 UTC period on 20 May and the 21:49:55–21:55:15 UTC period on 23 May are compared because of their similar temperature and $\chi^2_{\min} + \Delta\chi^2$ threshold used to determine the $(a,b)$ surfaces. Figure 16 illustrates the distribution of $\chi^2$ for the two periods, with the outlined region representing $\chi^2$ values that are $\leq 1$ for

the purpose of comparison. The region containing $\chi^2 \leq 1$ is 88.2% smaller for the 23 May flight leg compared to the 20 May period, and highlights how different $a$ and $b$ can yield a $\chi^2$ value that is within the given tolerance based on differences in the observed $TWC$ and $Z$ distributions. When bulk $TWC$ and $Z$ are compared against the 25 April (20 May) events, the median $Z$ from flight legs on 23 May is on average 34.4% (25.9%) lower while the median $TWC$ is 90.3% (43.9%) greater. As mentioned in Sect. 4, the sampling strategy on 23 May was different from the stratiform clouds observed with the previous two events

in that measurements were primarily made in the anvil region of supercell thunderstorms. Previous studies (e.g., Heymsfield et al., 2007) noted that the prefactor $a$ had less of a temperature dependence within anvil cirrus clouds, consistent with trends in $a$ for the 23 May flight legs.

### 6   Conclusions

This paper presented a novel approach to characterize the variability of mass-Dimension ($m$-$D$) coefficients characterizing

particle size distributions (PSDs) during the Mid-latitude Continental Convective Clouds Experiment (MC3E). The technique outlined here extends the approach of McFarquhar et al. (2015), who derived a volume of equally realizable solutions in the phase space of gamma fit parameter coefficients to characterize PSDs. Ground-based radar measurements of reflectivity $Z$ from the Vance Air Force Base, OK radar were matched to the location of the Cessna Citation II aircraft where total water content

(*TWC*) measurements from the Nevzorov probe were made and PSDs were derived from optical array probe data. These collocated datasets permitted use of a $\chi^2$ minimization technique where all $\chi^2$ within a tolerance $\Delta\chi^2$ of the minimum $\chi^2$ were considered equally plausible solutions to the $m = aD^b$ relationship for a flight leg of similar temperature. The tolerance was determined by considering uncertainties due to natural variability of cloud conditions for a particular environment, the statistical sampling of particles from the PSDs, and uncertainties in the measurements themselves.

The key findings of the paper are as follows:

1. The distribution of $\chi^2$ values in (*a,b*) phase space shows that the *a* and *b* parameters are highly correlated, as expected. The degree to which these $\chi^2$ values vary throughout a flight leg is influenced by how the PSDs, *TWC* from the Nevzorov probe, and *Z* from radar vary within a flight leg of similar temperature. Flight legs that have little variability in the microphysical properties and an allowable tolerance equal to the minimum $\chi^2$ in (*a,b*) phase space, such as the 10:03:05–10:08:45 UTC period on 25 April, occupy a surface area in (*a,b*) phase space that is up to 8.7 times larger than flight legs where microphysical properties vary more, such as the 11:05:20–11:14:45 UTC leg on the same day.

2. Surfaces of equally plausible solutions appear dependent on temperature for the 25 April and 20 May events. The range of plausible *a* and *b* coefficients is larger for flight legs of lower temperature, and 80% of the surfaces compared between the lowest and highest temperature for each day overlap by less than 50%.

3. Cases with little dependence of the surfaces of equally plausible solutions on temperature, like the flight legs analyzed on 23 May, can be explained in terms of the regions of cloud sampled and the types of ice hydrometeors observed. A mean overlap of 62.1% between four of the five (*a,b*) surfaces on that day is consistent with previous studies (e.g., Heymsfield et al., 2007) that note little dependence in the *a* coefficient with temperature in anvil cirrus clouds.

4. The minimum $\chi^2$ in (*a,b*) phase space determines the allowable tolerance $\Delta\chi^2$ for 5 of the 16 flight legs when determining the set of equally plausible *a* and *b* coefficients, whereas the combined uncertainty due to measurement error from the OAPs, Nevzorov *TWC* probe, and radar determines the $\Delta\chi^2$ for the remaining 11 flight legs. This means that the uncertainty in the *m-D* coefficients is driven by uncertainties in the measurements the majority of the time, with the natural parameter variability over a flight leg a driving factor for 31% of the flight legs observed. Thus, efforts to reduce measurement errors could reduce the uncertainty in derived (*a,b*) coefficients.

5. The covariability of *a* and *b* permit possible solutions of $b > 3$ for the ranges of particle sizes observed in 7 of the 16 flight legs analyzed. For these flight legs this covariability means that *Z* derived from *a* and *b* and the PSDs is still within 82.4% of the mean matched radar *Z*, which is marginally greater than the 50.5% difference when *b* is not greater than 3.

6. Flight legs where the cloud particles have lower effective density $\rho_e$, such as the -5.5 and -10.5 °C flight legs on 20 May, yield minimum *b* values in (*a,b*) phase space as much as 0.78 larger than clouds with a higher $\rho_e$ like the -16 and -23 °C legs on the same day. These differences can be explained by the different impacts of $\rho_e$ on *TWC* compared to *Z*.

A key finding of this study is that a range of $a$ and $b$ coefficients should be considered as equally plausible for a given environment due to the natural variability of cloud conditions and measurement uncertainties, even within a similar temperature range. This variability results in a large range of $a$ and $b$ as equally plausible solutions (as indicated in this study), and could explain the range in $m\text{-}D$ coefficients determined in past studies (Fig. 1) where $a$ coefficients can vary by 3 orders of magnitude and $b$ coefficients between 1 and 3 for measurements made in similar environmental conditions. The technique used in this study provides insight into how equally plausible $m\text{-}D$ coefficients can arise because the dependence of derived microphysical parameters on environmental conditions is sometimes more important than measurement uncertainties based on the instruments used to collect the data, but is not always the case. Further, it is shown that the dependence of the $(a,b)$ coefficients on temperature is still notable even when considering the ranges of equally plausible solutions. Future studies should further ascertain the extent to which the dependence of $(a,b)$ on other environmental parameters is robust enough to be distinguished from the natural variability of the surface or its variability due to measurement errors.

While representing $m\text{-}D$ coefficients as a range of equally plausible solutions may address shortcomings of microphysical parameterization schemes and remote sensing retrievals that employ a single $m\text{-}D$ relationship for a given ice species or environment, caution should be taken if the results presented here are applied to ranges of particle size or environments outside of those sampled (e.g., ones with different observed habits or various degrees of riming). The results presented here illustrate that similar $TWC$ and $Z$ can be obtained regardless of the $a$ and $b$ values chosen, with coefficients randomly selected from a surface of solutions allowing one to represent how the uncertainty in $(a,b)$ impacts any derived quantity. Thus, the large variability in derived $(a,b)$ for an equally plausible surface does not necessarily indicate there is a large uncertainty in quantities derived using the $a$ and $b$ coefficients. Future work should assess how the representation of modeled processes and retrieved quantities are influenced by the variability in $a$ and $b$ coefficients as well as which environmental drivers and cloud microphysical properties influence the size of derived surfaces of equally plausible solutions, and the extent to which measurement errors need to be reduced to better refine these surfaces. The approach presented in this study can be applied to additional studies that make use of collocated radar and microphysical measurements in other cloud and meteorological environments, and improve the statistical robustness of plausible $m\text{-}D$ parameters for given environmental conditions. Such studies may help to further understand how surfaces of equally plausible $(a,b)$ solutions are affected by different environments and the variability of cloud conditions therein, as well as the dependence of these solutions as a function of other cloud or environmental properties.

*Code and data availability.* The radar (doi: 10.5067/MC3E/NEXRAD/DATA202) and OAP (doi: 10.5067/GPMGV/MC3E/MULTIPLE/DATA201) data used in this study are found on the NASA GHRC MC3E data archive. The software packages used to match the radar data to the aircraft's location (https://github.com/swnesbitt/AWOT) and to process the OAP data (doi: 10.5281/zenodo.1285969) are openly available as GitHub repositories. The data containing matched radar and microphysical properties (doi: 10.13012/B2IDB-6396968_V1) used in this study are archived and available online.

# Appendix A: List of variables and their descriptions

$a$      Prefactor component in mass-Dimension relationship

$A$      Particle cross-sectional area

$b$      Exponent component in mass-Dimension relationship

$\chi^2$      Chi-square statistic for each $(a,b)$ over a flight leg

$\chi^2_{min}$      Lowest $\chi^2$ value in $(a,b)$ phase space for a flight leg

$\Delta\chi^2_1$      Threshold determined from uncertainty in the particle size distribution due to sampling statistics

$\Delta\chi^2_2$      Threshold determined from combined uncertainty due to measurement errors

$\Delta\chi^2$      Maximum value of $\chi^2_{min}$, $\Delta\chi^2_1$, or $\Delta\chi^2_2$

$D$      Particle maximum dimension

$D_{mm}$      Median mass diameter

$IWC$      Ice water content

$K_{DP}$      Specific differential phase

$|K_{\mathrm{ice}}|^2$      Dielectric constant for ice

$|K_w|^2$      Dielectric constant for water

$M(D)$      Mass distribution function

$N(D)$      Number distribution function

$N_t$      Total number concentration

$P$      Particle perimeter

$\rho_e$      Effective density

$SDP$      Sigma differential phase

$T$      Environmental temperature

$TWC$      Total water content measurement

$TWC_{\mathrm{diff}}$      Measure of normalized difference between the Nevzorov $TWC$ and that derived from the $N(D)$ for a given $(a,b)$ defined by Eq. (3)

$TWC_{\mathrm{SD}}$      $TWC$ derived from the $N(D)$ for a given $(a,b)$

$\zeta$      Particle sphericity

$Z$      Radar reflectivity factor

$Z_c(D)$      Cumulative reflectivity distribution function up to size $D'$

$Z(D)$      Reflectivity distribution function

$Z_{\mathrm{diff}}$      Measure of normalized difference between the radar $Z$ and that derived from the $N(D)$ for a given $(a,b)$ defined by Eq. (4)

$Z_{\mathrm{DR}}$      Differential reflectivity

$Z_{\mathrm{SD}}$      $Z$ derived from the $N(D)$ for a given $(a,b)$

$Z_t$      Derived total reflectivity from the mean $N(D)$ for a given $(a,b)$

*Author contributions.* JF prepared the manuscript and performed all the calculations with contributions from all co-authors. GM provided the idea and formulated the framework for the study, SN provided the framework for matching radar gates to an aircraft's position, WW processed particle data from the optical array probes, PZ conducted the radar bias calculations used in this study, and RR and HM provided feedback on the ideas and calculations presented.

5    *Competing interests.* The authors declare that they have no conflict of interest.

*Acknowledgements.* This research was supported by the US Department of Energy grants DE-SC0014065 and DE-SC0016476 (through UCAR subcontract SUBAWD000397), by the NASA Precipitation Measurement Missions grant NNX16AD80G, and by the National Science Foundation grant AGS-1247404. We thank all participants of MC3E for collecting the data used in this study.

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

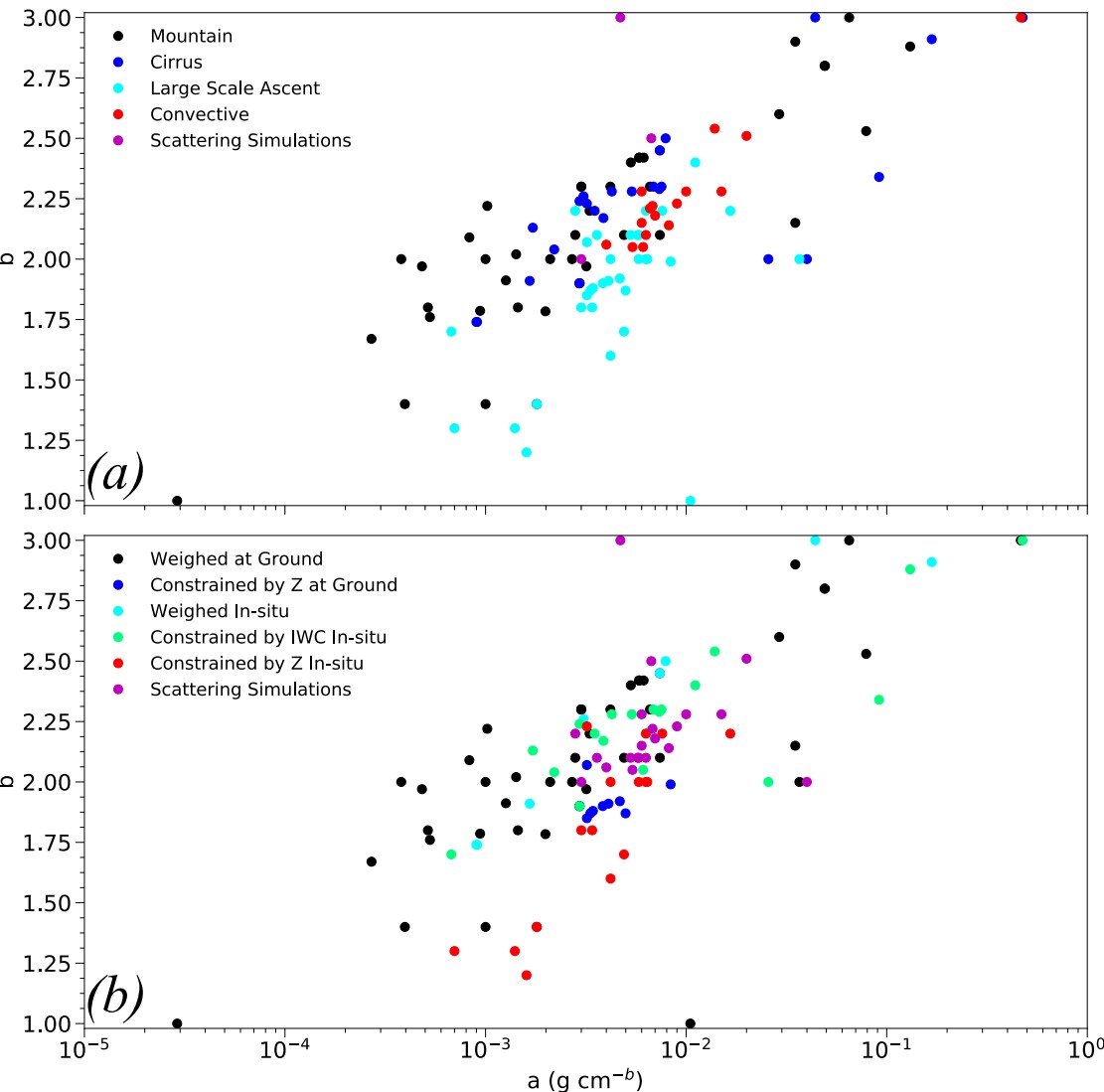

**Figure 1.** Distribution of *a* and *b* coefficients used to characterize $m = aD^b$ relationship from past studies. Points colored by the (a) environment in which measurements were made and (b) technique used to derive the relations.

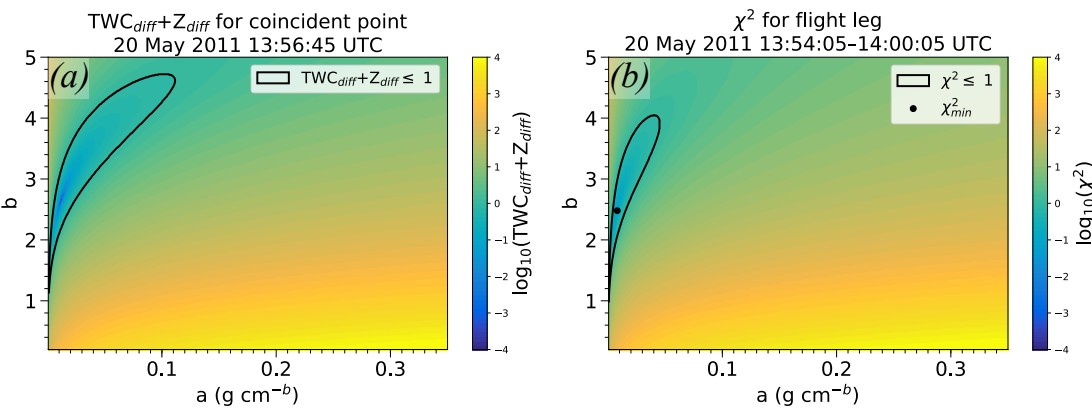

**Figure 2.** *TWC*$_{\text{diff}}$ + *Z*$_{\text{diff}}$ in (*a,b*) phase space for (a) a 10 s coincident point beginning 13:56:15 UTC on 20 May 2011 and (b) integrated over the encompassing flight leg between 13:54:14 and 13:59:35 UTC and normalized by the number of observations N. The black dot in (b) denotes the a and b minimizing $\chi^2$.

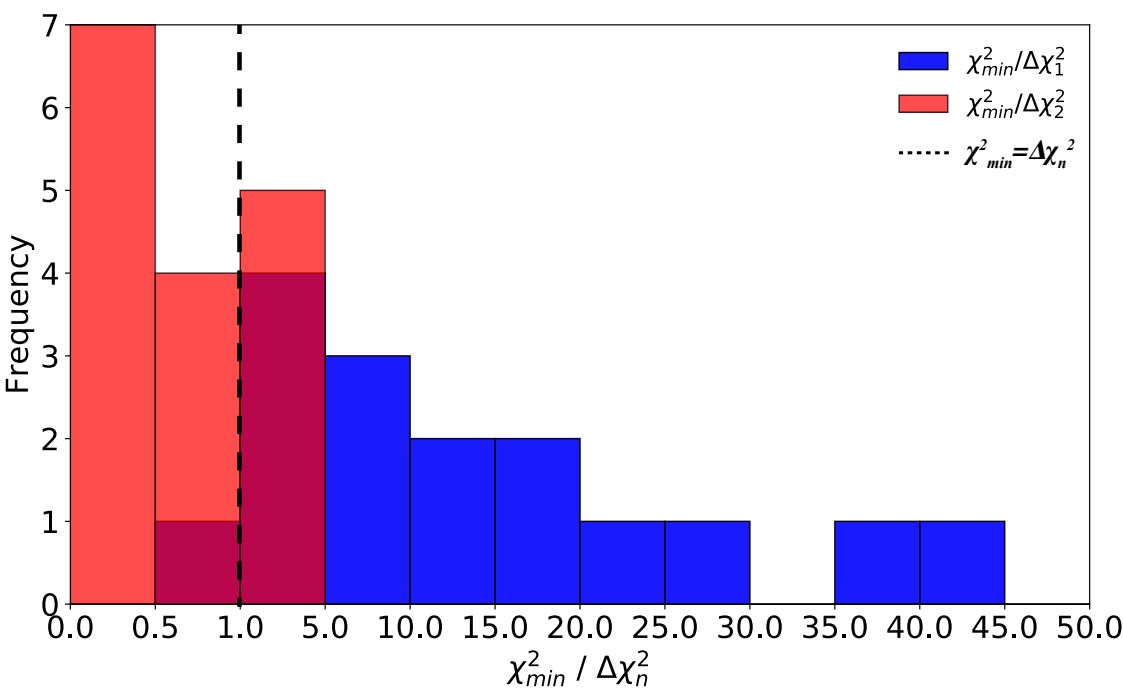

**Figure 3.** Frequency of $\chi^2_{min}/\Delta\chi^2_1$ (blue shading) and $\chi^2_{min}/\Delta\chi^2_2$ (red shading), where $\chi^2_{min}$, $\Delta\chi^2_1$, and $\Delta\chi^2_2$ derived for each flight leg used in analysis.

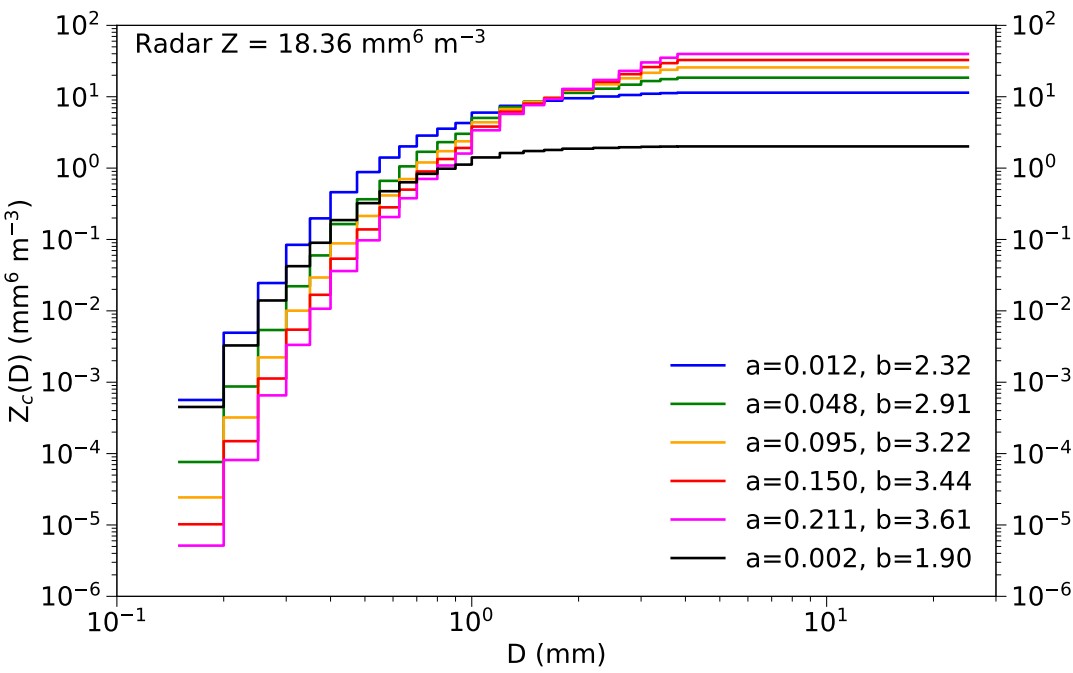

**Figure 4.** $Z_c(D)$ as a function of $D$ derived using modified $m$-$D$ coefficients from BF95 (black) and from the 5th (blue), 25th (green), 50th (orange), 75th (red), and 95th (magenta) percentiles from the set of equally plausible $m$-$D$ coefficients in order of increasing $b$ and $a$ values for the 14:16:30–14:32:15 UTC flight leg on 20 May 2011. Mean radar reflectivity matched at the aircraft's position for the same period is listed in top left.

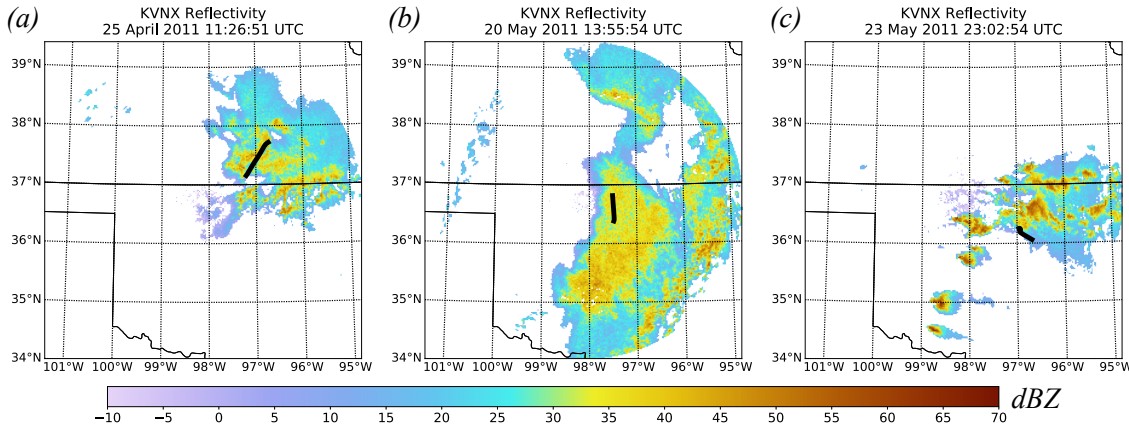

**Figure 5.** 0.5 degree PPI scan of corrected radar reflectivity from the KVNX radar for (a) 11:26:51 UTC 25 Apr 2011, (b) 14:04:34 UTC 20 May 2011, and (c) 23:02:54 UTC 23 May 2011. Black lines denote the Citation flight track for the constant-temperature leg corresponding to the radar image shown.

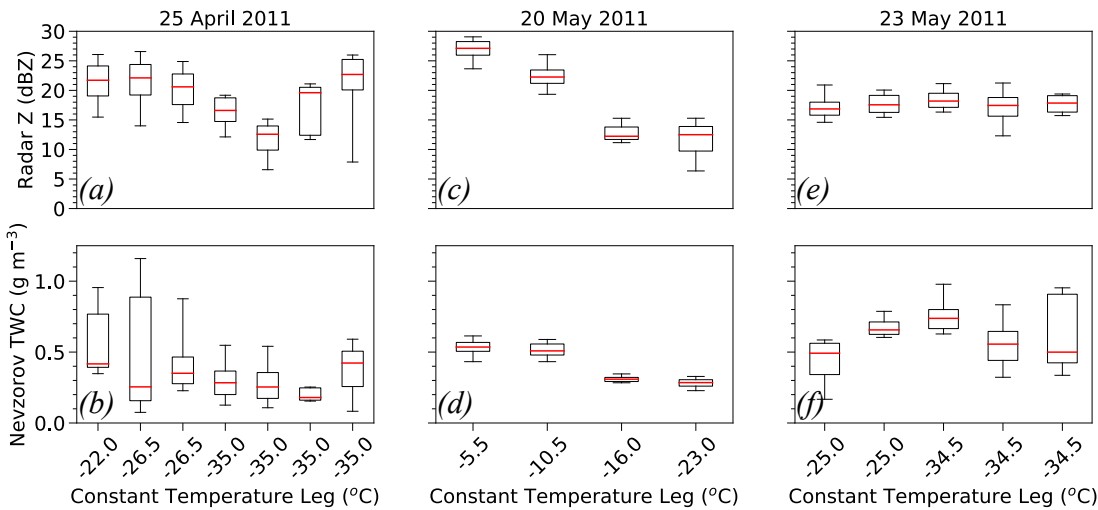

**Figure 6.** Distribution of matched *Z* (top) and *TWC* from the Nevzorov probe (bottom) for each constant-temperature leg on 25 Apr (left), 20 May (center), and 23 May 2011 (right). Whiskers represent the 5[th] and 95[th] percentiles, box edges are the 25[th] and 75[th] percentiles, and the line in the middle is the median. Cases where multiple legs of the same temperature exist are shown in chronological order.

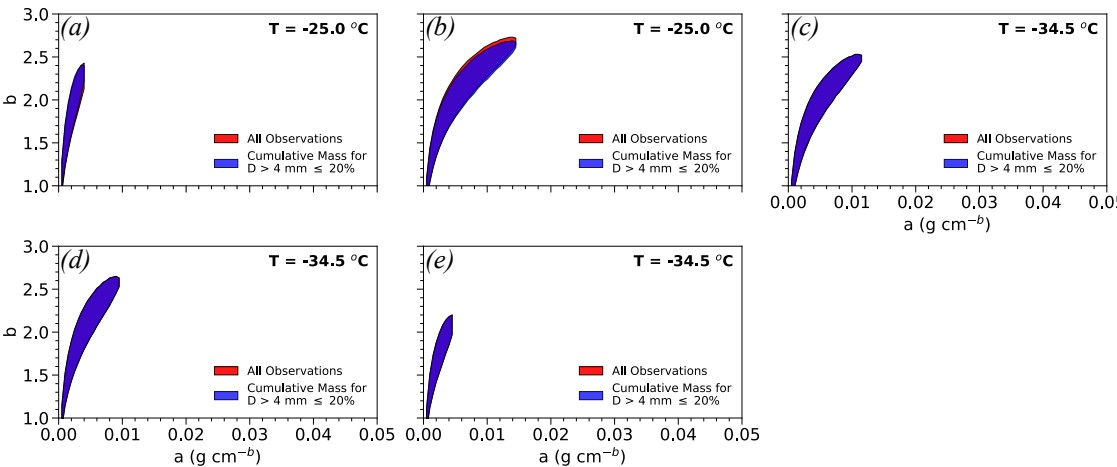

**Figure 7.** Surfaces of equally plausible $a$ and $b$ values from the $m = aD^b$ relation from each near-constant temperature leg on 23 May 2011 for all coincident observations (red) and only those where cumulative mass for $D > 4$ mm is $\leq 20$ % (blue). Flight legs of the same temperature are shown in chronological order.

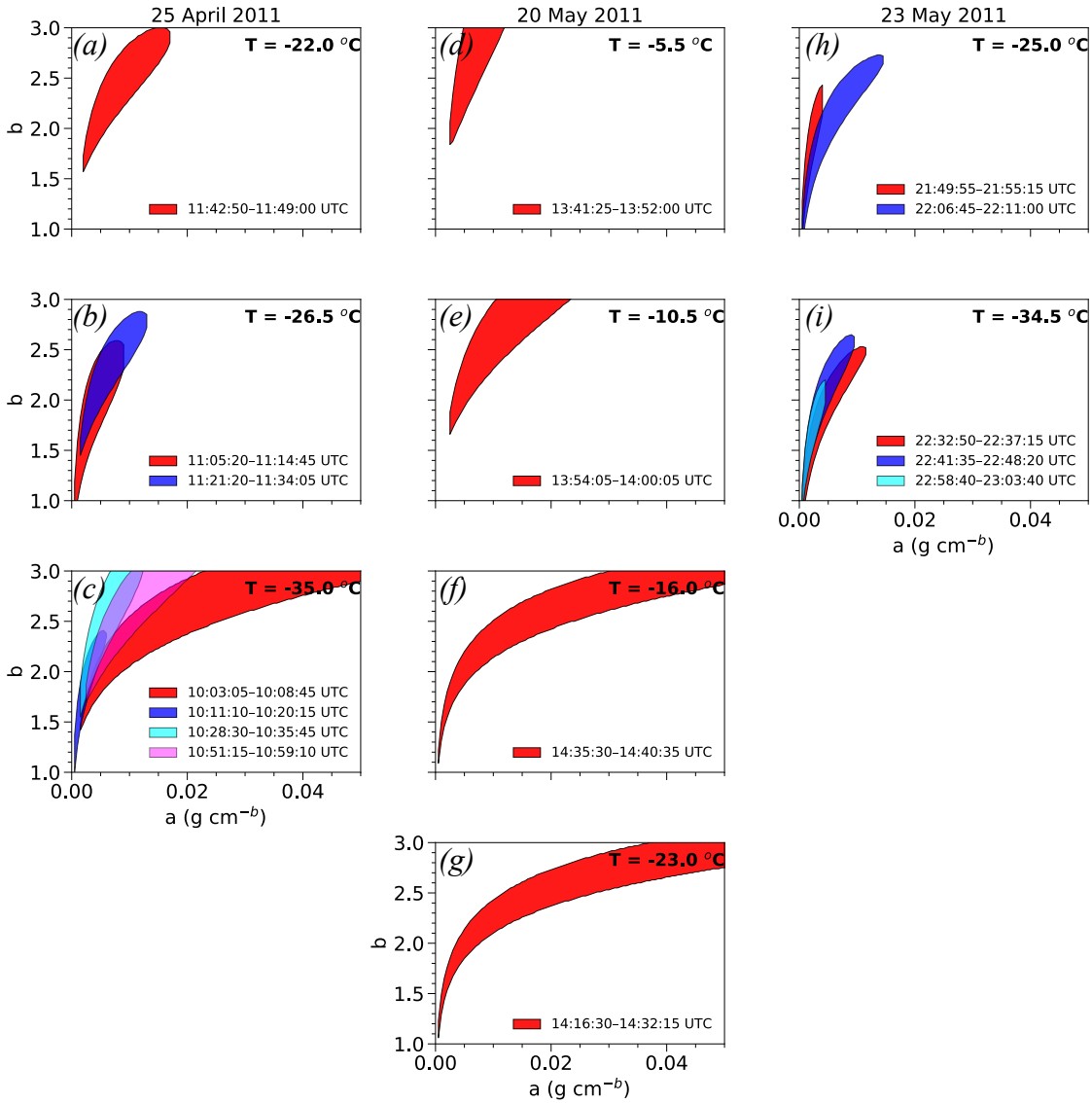

**Figure 8.** Surfaces of equally plausible *a* and *b* values for near-constant temperature flight legs for the (a–c) 25 April, (d–g) 20 May, and (h–i) 23 May 2011 events. Multiple legs occupying the same temperature are assigned a different color within a panel.

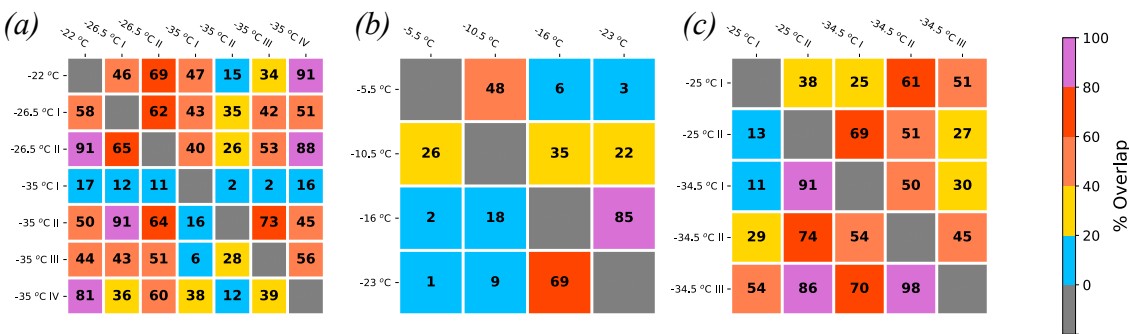

**Figure 9.** Matrix of overlap area between the equally plausible (*a,b*) surfaces corresponding to each row and column for (a) 25 April, (b) 20 May, and (c) 23 May 2011. The overlap area for each square is normalized by the area of the (*a,b*) surface corresponding to the flight leg listed in each row.

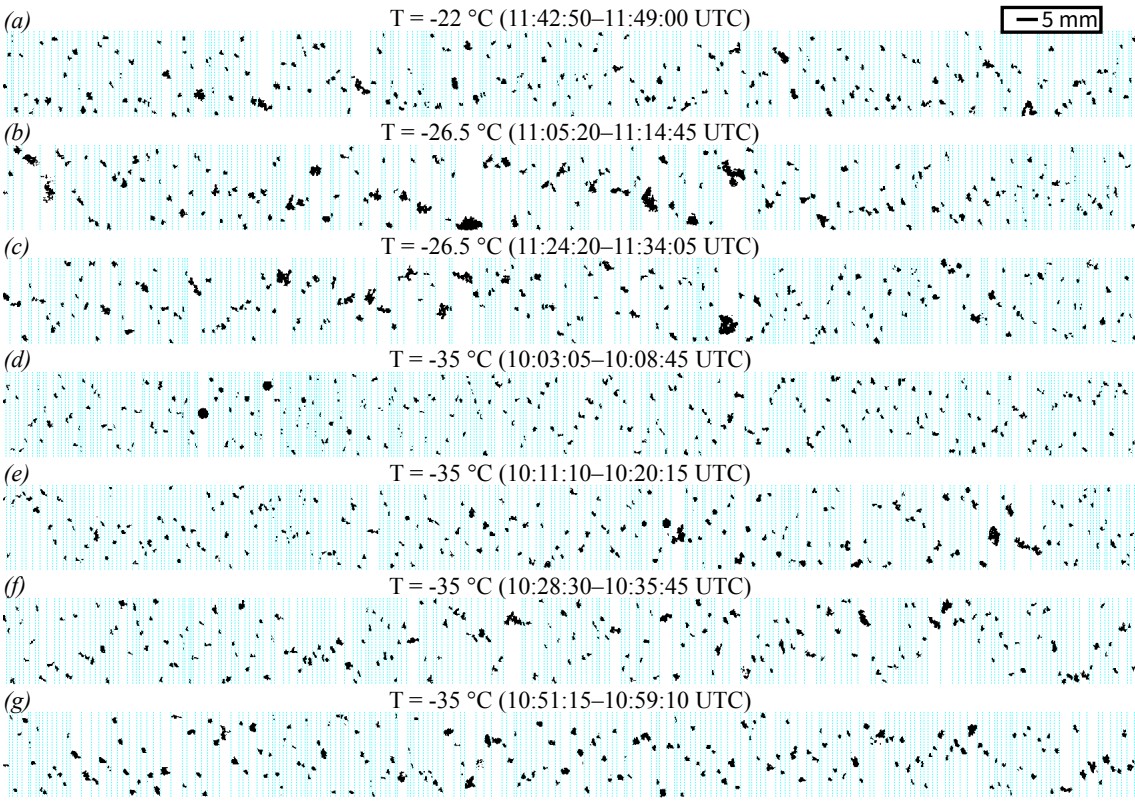

**Figure 10.** Representative particle images from the HVPS-3 for each near-constant temperature flight leg on 25 April 2011.

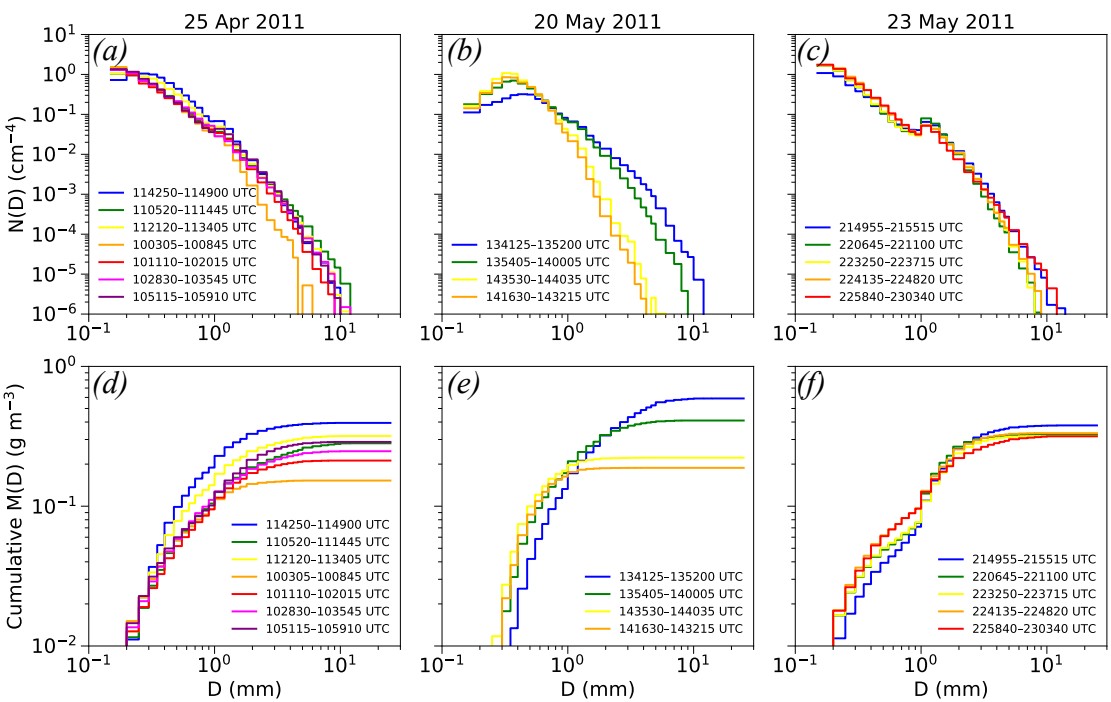

**Figure 11.** Mean *N(D)* (top) and cumulative *M(D)* (bottom) for each constant-temperature leg on 25 Apr (left), 20 May (center), and 23 May 2011 (right). Cases where multiple legs of the same temperature exist are shown in chronological order.

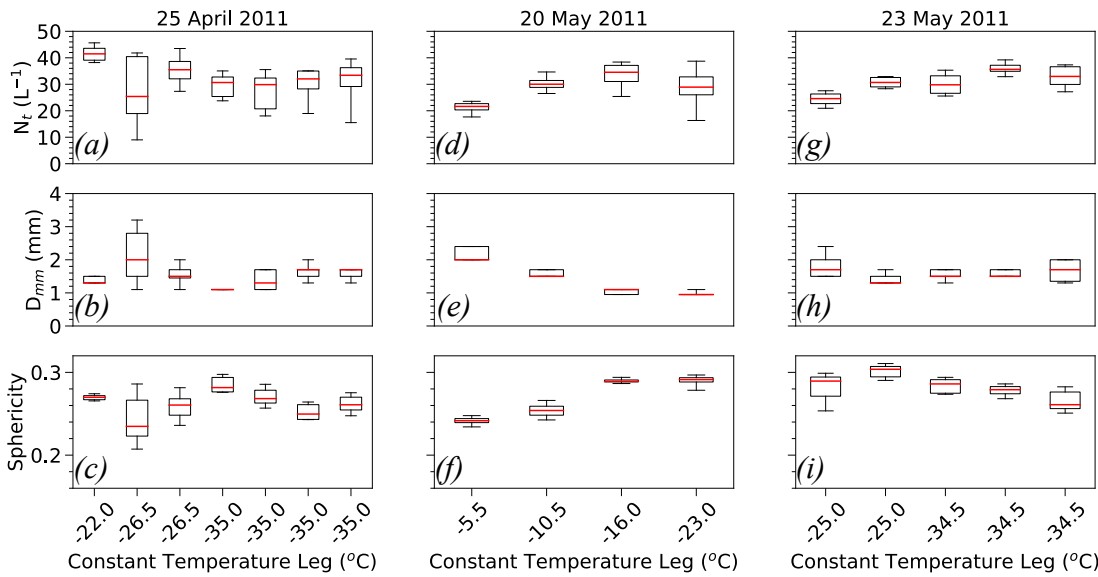

**Figure 12.** As in Fig. 6, but for number concentration $N_t$, median mass diameter $D_{mm}$, and mass-weighted mean sphericity.

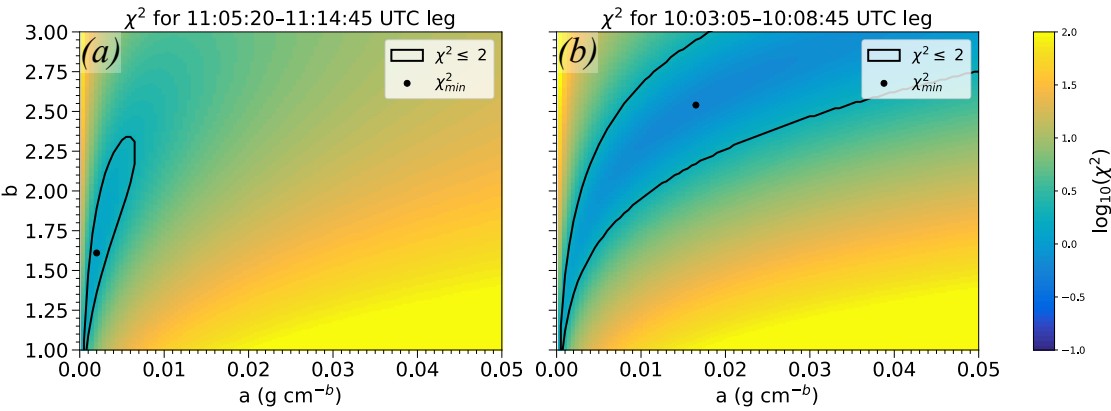

**Figure 13.** $\chi^2$ statistic in $(a,b)$ phase space for the (a) 11:05:20–11:14:45 UTC and (b) 10:03:05–10:08:45 UTC flight legs on 25 April 2011. Outlined regions represent $\chi^2 \leq 2$ and the dots $\chi^2_{min}$.

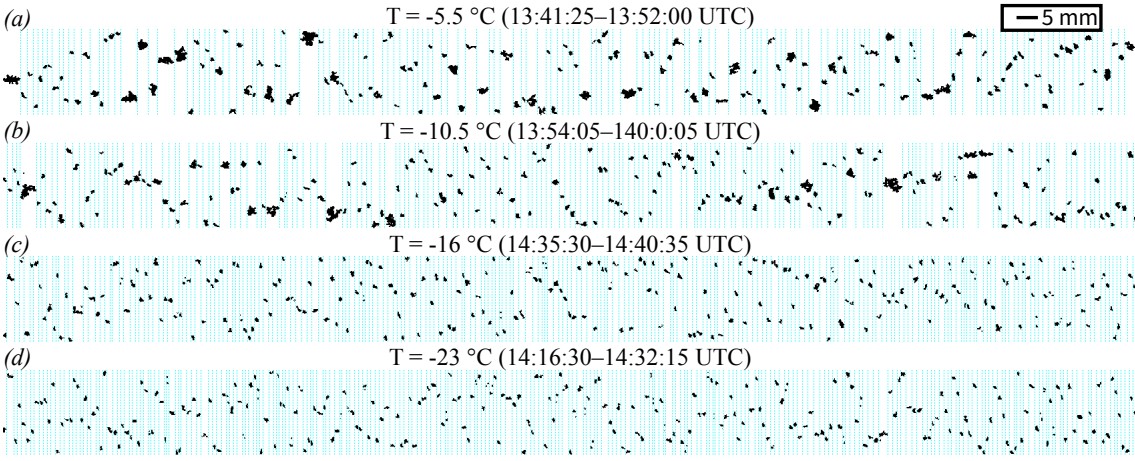

**Figure 14.** Same as in Fig. 10, but for the 20 May 2011 case.

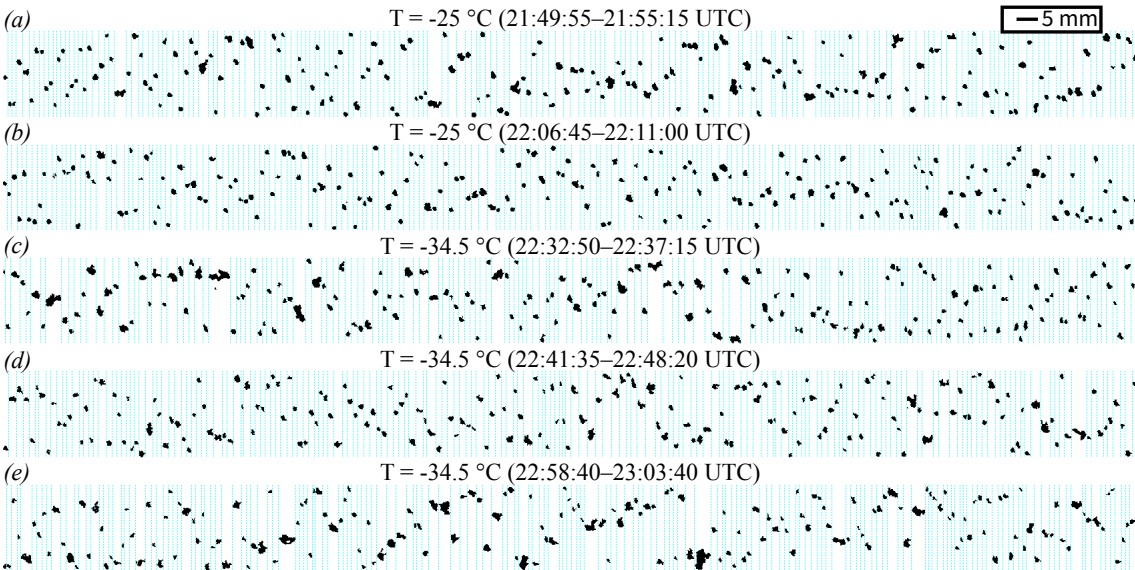

**Figure 15.** Same as in Fig. 10, but for the 23 May 2011 case.

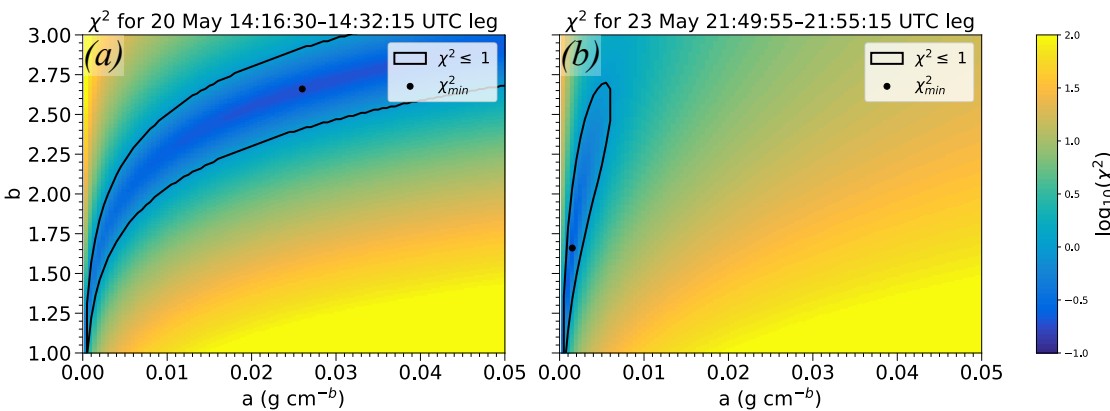

**Figure 16.** Same as in Fig. 13, but for (a) 14:16:30–14:32:15 UTC on 20 May and (b) 21:49:55–21:55:15 UTC on 23 May 2011. Outlined regions represent $\chi^2 \leq 1$ and the dots $\chi^2_{\min}$.

**Table 1.** List of constant temperature flight legs used in the analysis for which coincident data between the ground-based radar and UND Citation exist. Start and end times, mean altitude, and temperature displayed.

| Mean Temp. [°C] | Mean Alt. [km] | Start Time [UTC] | End Time [UTC] |
|---|---|---|---|
| 25 April 2011 | | | |
| -22.0 | 6.8 | 11:42:50 | 11:49:00 |
| -26.5 | 7.4 | 11:05:20 | 11:14:45 |
| -26.5 | 7.4 | 11:21:20 | 11:34:05 |
| -35.5 | 8.3 | 10:03:05 | 10:08:45 |
| -35.5 | 8.3 | 10:11:10 | 10:20:15 |
| -35.5 | 8.3 | 10:28:30 | 10:35:45 |
| -35.5 | 8.3 | 10:51:15 | 10:59:10 |
| 20 May 2011 | | | |
| -5.5 | 5.0 | 13:41:25 | 13:52:00 |
| -10.5 | 5.9 | 13:54:05 | 14:00:05 |
| -16.0 | 6.9 | 14:35:30 | 14:40:35 |
| -23.0 | 7.9 | 14:16:30 | 14:32:15 |
| 23 May 2011 | | | |
| -25.0 | 7.9 | 21:49:55 | 21:55:15 |
| -25.0 | 7.9 | 22:06:45 | 22:11:00 |
| -34.5 | 9.1 | 22:32:50 | 22:37:15 |
| -34.5 | 9.1 | 22:41:35 | 22:48:20 |
| -34.5 | 9.1 | 22:58:40 | 23:03:40 |

**Table 2.** List of constant temperature flight legs and the ratio between $Z_{\text{diff}}$ and $TWC_{\text{diff}}$ valid at the $(a,b)$ that minimize $\chi^2$.

| 25 April 2011 | | 20 May 2011 | | 23 May 2011 | |
|---|---|---|---|---|---|
| Times [UTC] | $\frac{Z_{\text{diff}}}{TWC_{\text{diff}}}$ | Times [UTC] | $\frac{Z_{\text{diff}}}{TWC_{\text{diff}}}$ | Times [UTC] | $\frac{Z_{\text{diff}}}{TWC_{\text{diff}}}$ |
| 11:42:50–11:49:00 | 2.02 | 13:41:25–13:52:00 | 4.92 | 21:49:55–21:55:15 | 1.52 |
| 11:05:20–11:14:45 | 0.81 | 13:54:05–14:00:05 | 6.31 | 22:06:45–22:11:00 | 1.82 |
| 11:21:20–11:34:05 | 1.62 | 14:35:30–14:40:35 | 3.2 | 22:32:50–22:37:15 | 0.99 |
| 10:03:05–10:08:45 | 0.8 | 14:16:30–14:32:15 | 3.99 | 22:41:35–22:48:20 | 1.82 |
| 10:11:10–10:20:15 | 1.5 | | | 22:58:40–23:03:40 | 0.32 |
| 10:28:30–10:35:45 | 8.58 | | | | |
| 10:51:15–10:59:10 | 1.76 | | | | |