# Peer review of "A novel approach to characterize the variability in mass-Dimension relationships: results from MC3E"

_Atmospheric Chemistry and Physics, 2018_

## Referee Comment (RC1) · D. Baumgardner (Referee) · 28 Sep 2018

The statistical analysis that is conducted in this study is thorough in many regards, lacking clarity in some places, and requires some further discussion and analysis yet in other places. The general premise of the study is that further evaluation is needed of the mass-diameter equations that are used to derive cloud water from remote sensing measurements. This is an extension of the analysis that has been done by others, and is amply referenced in the introduction. Although I consider this a useful study, worth of publication, I am reluctant to immediately recommend it for publication without having my primary concerns addressed, as well as some of my lesser ones that are listed

after I enumerate the more important questions (to me). 1. The methodology by which TWC and reflectivity (Z) are derived from the size distributions need to be explicitly stated, in equation form, at the first of the paper. TWC is never shown and Z is not shown until the discussion of how the equally plausible surfaces (EPS) are computed. It seems more logical to have the equations on TWC and Z prior to the error term rather than afterwards. 2. When the computation of TWC and Z is introduced, there has to be a more vigorous discussion of the expected uncertainties when deriving the size and the mass from the 2D images. This is also why it becomes confusing at a later point when there is a discussion of the effective density and its impact on the EPS. What effective densities were used to derive TWC in the first place? Wouldn't that bound the uncertainty in the a&b coefficients? 3. How was the 10 second averaging period derived? Wouldn't it have been much more consistent to use variable sampling periods that always ensured statistically significant number of particles? 4. What is the rational for equally weighting the TWC and Z error terms? To me, this is a significant assumption that needs a more thorough discussion. The Nevzorov TWC probe has a sample area much smaller than the 2D and HVPS while all three instruments sample a volume many orders of magnitude smaller than the radar. How do you reconcile these differences? If you construct the EPS from the TWC and Z independently, are they similar, or does TWC drive the minimization some of the time and the Z others? I understand, an appreciate, the care that was taken to obtain homogeneous samples from the Z data, but ask that my question be addressed in the manner I suggest. 5. I found the summary somewhat incomplete in that it concludes that there will always be a broad region of EPS in any given situation. If I am a radar or satellite scientist, this would lead me to throw up my hands, put the EPS chart on the door, and throw a dart. Is this what the authors suggest we do? If not, then I strongly suggest that the paper end on a more positive note that can recommend to the remote sensing community what should be done. Less major comments (that still require attention) 1. Why was the CIP not used? It is introduce as one of the probes on the aircraft, it has twice the number of diodes as the 2D-C, 100 mm between arms rather than 63, and hence

more sample volume, yet it isn't used. Why? 2. How are the 2D-C and HVPS size distributions combined? Do they always overlap well? If not, how is this reconciled? 3. Error analysis, error analysis, error analysis. What are the expected uncertainties in the EPS due to choice of size and effective density? 4. Page 5, line 17. "Following Heymsfield and Baumgardner (1985) and Field (1999), only particles with a center of mass within the OAP's field of view were considered as otherwise there is too much uncertainty in particle shape.". Several corrections/questions here. First of all, it it the center of mass being located in the field of view or center of the measured image? In either case, how is this determined? Finally, using the "center-in" technique mostly reduces uncertainty in size, not in shape. 5. Equation 3 shows TWC being averaged not Z, but don't the derived Zs also get averaged?

---

## Referee Comment (RC2) · Anonymous Referee #2 · 12 Nov 2018

General comment

The manuscript presents a new method developed to characterize the variability of m-D power law relationship coefficients a and b. The technique minimizes the chi-square difference between TWC and Z derived from a combination of OAPs and corresponding TWC and Z directly measured by a Nevzorov probe and an S band radar. All a and b within a specified tolerance were regarded as equally plausible solutions (EPS). A chosen chi-square criteria (though questionable) is used to produce most likely solutions (minimization of chi-square in a-b contour plot) for a and b out of all EPS.

Overall, the presented method is smart with potential to retrieve adequate and most

likely a and b coefficients for whatever specific data set, subset, temperature range, stratiform / convective cloud region, cloud type, etc...

However, instrumental shortcomings (S band radar, 2D-C with 32 photo-diodesof $25\mu$m resolution?) and uncertainties need to be considered. Also authors are facing poor data statistics (used data < 1.5 hours of sampling with flight legs between -5°C and -35°C, only three cloud systems, with 20 May cloud not comparable in temperature to two other clouds ) due to necessary co-location with ground radar. The lack of data statistics does not allow to assess reliable results for underlying data set. The results of most likely a and b coefficients for this limited data set may not be statistically well grounded to be used for model and remote sensing applications.

The question arises, if it wouldn't be better to more precisely present the method, even refine the method, where mathematical basis is missing, then applying the method to a limited dataset, just to demonstrate how the method works, without having to deal with uncoherent results due to lack of statistically valuable data. An idea could be to publish solely the method with application illustrations (AMTD ?).

Main major points

Method : - Manuscript needs to document equation illustrating how Z is calculated from PSD. - The authors have to quantify and discuss S band radar reflectivity factor sensitivity as a function of crystal size. - Don't use B&F for MMD_max calculation (Fig 11, etc... and respective arguments in text). B&F has been retrieved for mean chord length size definition and therefore would necessarily overestimate TWC, real MMD_max, etc...when using PSD in D_max definition. - Either the manuscript has to derive mathematically how TWC_diff (equ 1) and Z_diff (equ. 2) can be merged into a sum or likewise into chi-square (equ.3) both to be minimized subsequently (Fig 2). As it stands, we feel that differences (measured, calculated from PSD) of the square root of Z may be a bit comparable to respective differences in TWC, in order to merge those two terms. Of course this is scientifically insufficient... The authors may think about

sensitivity studies for table 1 data to know which term controls chi-square (equ 3), as a function of different flight legs and how this evolves during individual flight legs. - Poisson statistics doesn't take into account systematic uncertainties in measuring TWC from Nevzorov and not systematic errors in Z measurements. Within a homogeneous cloud segment the statistical counting uncertainty is small, but systematic measurement uncertainty can be of the order of 50% or 100 % …. This has to be taken into account instead of Poisson statistics that seems serving to get rid of the uncertainty discussion… ? To resume, the tolerance of EPS has to take into account real uncertainties and not just Poisson statistics / natural variability. - Recommendation to limit b to 1-3. As a consequence delete everything from pg 7 line 20 to pg 8 line 21, since this discussion finally does not contribute to the study. I don't see the physical meaning to go beyond b=3 of a sphere, mathematically of course one may not care about an interpretation. - - Data / instrumentation / uncertainty consideration - Use PSD number uncertainty of 50% for larger particles and calculate associated TWC_SD uncertainty. - Likewise Z_SD uncertainty from Z equation that you have to present. See comment above. - How supercooled water has been quantified? Excluded from analysis? Be aware of the fact that Nevzorov will not correctly quantify LWC when IWC is more or less dominating TWC. - Please document average number and mass PSD (additional figure!) for table 1 flight sequences. - Nevzorov TWC uncertainty: In literature several times has been documented that 2DC+2DP is matching the shallow Nevzorov TWC. Likewise for the deep cone in other publications. This illustrates that a and b coefficients can be adapted to match TWC from instruments, but it does not prove that shallow or deep cone Nevzorov collect ice at 100%. May be something between 50-100% for the deep cone. Check literature. Thus, the manuscript has to take into account possible systematic uncertainties of PSD number, Nevzorov TWC, and also measured Z impacting the tolerance of EPS and finally guiding recommendations of most likely a and b coefficients. …. - Figure 9: if the underlying dataset would be statistically a little more representative (which is certainly not the case) wouldn't we expect more organized colours (contour plot gradient) perpendicular to the diagonal line of the

grey matrix elements? - 20 May flight: these are solely 'low altitude' data with T>-23, whereas for the other two flights used in this study, temperature T<-22. What is the argument to choose the 20 May flight since limited comparison with colder temperature data of two other flights? -

Further Comments

1. Pg 1, line 18: It does not make sense to establish m-D relation in mixed phase clouds. We know m-D for water droplets. You should exclude all mixed phase sequences from data! 2. Figure 1 is not thoroughly documented with literature references and thus not reproducible. 3. Pg 4, line 14: redo analysis with CDP droplet probe not exceeding 10 cm-3 on a 1 s basis may be a more suitable approach. 4. Pg 4, line 18 ff: Assuming that S band radar reflectivity is not significant for sub-millimetric particles, what is the TWC percentile at 1mm of the cumulative mass PSD ? Please don't use B&F but for example Heymsfield 2010, etc.. retrieved for D_max definition. 5. Pg 8, line 14: 85 cm? 6. Pg 9, line 26: Use m-D relation, for mass content estimation, other than BF95 for D_max size definition. 7. Pg 11, line 5; Fig 11: Idem 8. Pg 15, line 15 idem 9. Pg 11, line 12; You mean just Fig 11 in this sentence? 10. Pg 11, equation 6: How sphericity results deviate when comparing two instruments of different pixel resolution having sampled the same paricle? And how the averaging of sphericity has been performed over all crystals of a crystal population? For OAP 2D images the sphericity is size dependent, partly due to bias of 2D projections sorted into size classes, with larger particles having smaller 2D image sphericity than smaller particles. 11. Pg 12 line 16: Why applying factor 0.6 in reference volume for retrieving effective density, impacting TWC???

---

## Author Comment (AC1) · 25 Jan 2019

**Review Responses for "A novel approach to characterize the variability in mass-Dimension relationships: results from MC3E"**

We thank the reviewers for their careful reviews and constructive critique of our paper. We feel that responding to their comments has considerably improved the quality of the manuscript. In our response below, the comments of the reviewers are listed first, our response second, and changes to the text third. Page and line references in our response apply to a version of the revised manuscript that contains track changes

and is included as a supplement, with added text wavy-underlined and in blue and discarded text struck out and in red.

**Response to Comments of Reviewer 1:**

**Major comment #1:** The methodology by which *TWC* and reflectivity (Z) are derived from the size distributions need to be explicitly stated, in equation form, at the first of the paper. *TWC* is never shown and *Z* is not shown until the discussion of how the equally plausible surfaces (EPS) are computed. It seems more logical to have the equations on *TWC* and *Z* prior to the error term rather than afterwards.

**Response:** An explanation of how *TWC* and *Z* was derived from the PSDs, including addition of Eq. (1) and Eq. (2) that define $TWC_{SD}$ and $Z_{SD}$, has been added to line 27 on page 6.

**Text:** For an individual 10 s sample, the *TWC* and *Z* derived from the PSD for a specific *a* and *b* is given by $TWC_{SD}$ and $Z_{SD}$, respectively, as

$$TWC_{SD} = \sum_{j=1}^{N}(aD^b)N(D_j)dD_j \text{ and} \tag{1}$$

$$Z_{SD} = \left(\frac{6}{\pi\rho_{ice}}\right)\frac{|K_{ice}|^2}{|K_w|^2}\sum_{j=1}^{N}(aD^b)^2N(D_j)dD_j \tag{2}$$

following the method of Hogan et al. (2006) and accounting for the different dielectric constants for water ($|K_w|^2$ = 0.93) and ice ($|K_{ice}|^2$ = 0.17). Uncertainties in the $TWC_{SD}$ and $Z_{SD}$ are discussed later in this section.

**Major comment #2:** When the computation of *TWC* and *Z* is introduced, there has to be a more vigorous discussion of the expected uncertainties when deriving the size
and the mass from the 2D images. This is also why it becomes confusing at a later point when there is a discussion of the effective density and its impact on the EPS. What effective densities were used to derive *TWC* in the first place? Wouldn't that bound the uncertainty in the a&b coefficients?

**Response:** Potential uncertainties of $TWC_{SD}$ and $Z_{SD}$ are first acknowledged in line 25 on page 6, and are now numerically defined in the manuscript. Please see our response to Reviewer 2's major comment #5 for a full response and additions to the manuscript regarding measurement uncertainties from the OAPs, Nevzorov probe, and radar. As for how these uncertainties bound the *a* and *b* coefficients, this is now discussed in the manuscript on line 22 of page 8.

**Major comment #3:** How was the 10 second averaging period derived? Wouldn't it have been much more consistent to use variable sampling periods that always ensured statistically significant number of particles?

**Response:** Text has been added on page 4, line 13 to clarify that start and end times of each 10 s sampling periods were determined such that radar echo and microphysical data continuously existed throughout each 10 s period, thus ensuring that analysis did not include periods where the aircraft may be entering/exiting cloud. Further, points with the mean *TWC* for each 10 s period $< 0.05$ g m$^{-3}$ were ignored (lines 17–18, page 4). Based on some of our previous work (e.g., McFarquhar et al., 2007 MWR), this suggests there are statistically significant numbers of particles for the analyzed time periods.

**Text:** Each 10 s period determined required radar echo and microphysical data for all 1 s samples to ensure that the aircraft and matched radar *Z* were completely in cloud during the 10 s period.

**Text:** Observations where the mean *TWC* for a 10 s interval $< 0.05$ g m$^{-3}$ were ignored as the values were considered either below the noise threshold of the Nevzorov probe

[Figure]

or optically thin cloud.

**Major comment #4:** What is the rational for equally weighting the *TWC* and *Z* error terms? To me, this is a significant assumption that needs a more thorough discussion. The Nevzorov *TWC* probe has a sample area much smaller than the 2D and HVPS while all three instruments sample a volume many orders of magnitude smaller than the radar. How do you reconcile these differences? If you construct the EPS from the *TWC* and *Z* independently, are they similar, or does *TWC* drive the minimization some of the time and the *Z* others? I understand, an appreciate, the care that was taken to obtain homogeneous samples from the *Z* data, but ask that my question be addressed in the manner I suggest.

**Response:** In general, we wanted the *TWC* and *Z* error terms to have approximately equal weight so that different portions of the measured size distribution would each be having some impact on the derived *a/b* coefficients (if we weighted according to the sample volume, this would be almost equivalent to not using any information from the bulk content probe). Discussion has been added to lines 12–21 on page 7 regarding how $Z_{diff}$ was weighted using a priori assumptions of *Z* being proportional to the square of a particle's mass. Since cloud conditions often impact *Z* differently than TWC, $TWC_{diff}$ has a greater impact on the $\chi^2$ minimization technique some of the time and $Z_{diff}$ has a greater impact on the minimization other times. Table 2 has been added to outline the relative difference between $Z_{diff}$ and $TWC_{diff}$ for each flight leg used in the study. $TWC_{diff}$ and $Z_{diff}$ were ultimately not adjusted to be given equal weight for each flight leg as that would have diminished the utility of using both *TWC* and *Z* observations, and limit the ability to diagnose periods when cloud properties influence *TWC* and *Z* differently.

**Text:** Given a priori assumptions of *Z* being proportional to the square of a particle's mass, the square root of reflectivity was used in Eq. (4) so that $TWC_{diff}$ would be similar to $Z_{diff}$ on average and each would have approximately equal weight in determining

[Figure]

*a* and *b*. Although radar *Z* measurements involve a significantly greater sample volume than that of OAPs and a bulk content probe, $TWC_{diff}$ and $Z_{diff}$ were not weighted proportionally to the sample volume in order to ensure that both bulk moments had some impact on the derived *a* and *b*. Given that larger ice crystals are fractionally more important than small crystals in determining $Z_{SD}$ than $TWC_{SD}$ and given varying contributions of larger crystals to $Z_{SD}$ and $TWC_{SD}$, $TWC_{diff}$ has a greater impact on the $\chi^2$ minimization procedure some of the time while $Z_{diff}$ does at other times. The ratios between $Z_{diff}$ and $TWC_{diff}$ for each flight leg are given in Table 2, and range between 0.32 and 8.58 with a mean of 2.62 among the 16 flight legs. No attempt is made to force equal weight for $Z_{diff}$ and $TWC_{diff}$ for each coincident point because there are periods when cloud properties influence *TWC* differently than Z.

**Major comment #5:** I found the summary somewhat incomplete in that it concludes that there will always be a broad region of EPS in any given situation. If I am a radar or satellite scientist, this would lead me to throw up my hands, put the EPS chart on the door, and throw a dart. Is this what the authors suggest we do? If not, then I strongly suggest that the paper end on a more positive note that can recommend to the remote sensing community what should be done.

**Response:** Discussion has been added on line 28 of page 17 that emphasizes how the sizes of the EPS are related to the correlation between *a* and *b* and that the large variability in (*a,b*) does not necessarily relate to large uncertainties in all derived quantities. Suggestion of future work aimed at assessing the drivers behind the size of the EPS and also how the EPS can be applied within numerical models and remote sensing retrievals has also been added to the text.

**Text:** The results presented here illustrate that similar *TWC* and *Z* can be obtained regardless of the *a* and *b* values chosen, with coefficients randomly selected from a surface of solutions allowing one to represent how the uncertainty in (*a,b*) impacts any derived quantity. Thus, the large variability in derived (*a,b*) for an equally plausible

surface does not necessarily indicate there is a large uncertainty in quantities derived using the *a* and *b* coefficients. Future work should assess how the representation of modeled processes and retrieved quantities are influenced by the variability in *a* and *b* coefficients as well as which environmental drivers and cloud microphysical properties influence the size of derived surfaces of equally plausible solutions, and the extent to which measurement errors need to be reduced to better refine these surfaces.

**Minor comment #1:** Why was the CIP not used? It is introduced as one of the probes on the aircraft, it has twice the number of diodes as the 2D-C, 100 mm between arms rather than 63, and hence more sample volume, yet it isn't used. Why?

**Response:** Text has been added to lines 16–25 on page 5 acknowledging the greater sample volume of the CIP, and an explanation of why the 2D-C having anti-shattering tips and very little indication of shattered artifacts from inter-arrival time distributions provided an advantage over the CIP for this particular study as explained in Wu and McFarquhar (2016). Closer examination of that manuscript and of the size distribution code used to produce the PSDs prompted removing text on page 5, line 21 of the original manuscript that mentioned use of an inter-arrival time threshold for the 2D-C.

**Text:** The 2D-C was used instead of the CIP in the analysis even though the CIP has a larger sample volume because the inclusion of anti-shattering tips on the 2D-C reduced the impact of shattered artifacts (e.g., Korolev et al., 2011). Previous studies (Korolev et al., 2011, 2013a; Jackson et al., 2014) have shown that use of algorithms to identify shattered artifacts are sometimes needed even when the OAP is equipped with anti-shattering tips. Artifacts are identified by examining the frequency distribution of the times between which particles enter the sample volume (inter-arrival time; Field et al., 2006). When artifacts are present, this distribution follows a bimodal distribution with naturally-occurring particles having a mode with longer inter-arrival times and shattered artifacts having a mode with shorter inter-arrival times (e.g., Field et al., 2003). During MC3E there was only one mode in the inter-arrival time distribution corresponding to

the naturally-occurring particles (Wu and McFarquhar, 2016) at all times, suggesting there were few shattered artifacts. Therefore, no shattering removal algorithm was used for the 2D-C and HVPS.

**Minor comment #2:** How are the 2D-C and HVPS size distributions combined? Do they always overlap well? If not, how is this reconciled?

**Response:** While some studies have used variable weighting within the overlap region to produce a smoother transition in *N(D)* between two OAPs (e.g., Fontaine et al., 2014 ACP), the small discrepancy in *N(D)* within the overlap region (5%) characterized for the MC3E campaign justified use of a single, consistent cutoff of 1 mm for Wu and McFarquhar (2016) and this study. Text has been added on line 26 of page 5 to clarify this point.

**Text:** The 1 mm cutoff was chosen since *N(D)* for the two OAPs agreed on average within 5 percent for $0.8 \leq D \leq 1.2$ mm, and was used for all PSDs irrespective of periods when the difference between *N(D)* for the OAPs exceeded 5% in the overlap region.

**Minor comment #3:** Error analysis, error analysis, error analysis. What are the expected uncertainties in the EPS due to choice of size and effective density?

**Response:** The derived EPS now account for uncertainties due to measurement error from the OAPs, Nevzorov *TWC* probe, and the radar reflectivity. Please see our response to Reviewer 2's major comment #5 for a full response addressing these measurement errors.

**Minor comment #4:** Page 5, line 17. "Following Heymsfield and Baumgardner (1985) and Field (1999), only particles with a center of mass within the OAP's field of view were considered as otherwise there is too much uncertainty in particle shape.". Several

corrections/questions here. First of all, it it the center of mass being located in the field of view or center of the measured image? In either case, how is this determined? Finally, using the "center-in" technique mostly reduces uncertainty in size, not in shape.

**Response:** The text on lines 1–8 of page 6 have been modified to clarify the points above, and to briefly describe the criterion used to determine whether a particle was "center-in".

**Text:** Following Heymsfield and Baumgardner (1985) and Field (1999), only particles imaged with their center within the OAP's field of view were considered as otherwise there is too much uncertainty in particle size. Particles were identified as having their center within the field of view if their maximum dimension along the time direction exceeded the largest length where the particle potentially touched the edge of the photodiode array.

**Minor comment #5:** Equation 3 shows *TWC* being averaged not Z, but don't the derived Zs also get averaged?

**Response:** The reviewer is correct in that reflectivity is also averaged in the $\chi^2$ calculation. This was properly computed for the study, and a pair of brackets has been placed around the $TWC_{diff}$(i) + $Z_{diff}$(i) term of Eq. (3) (now Eq. (5) in the revised manuscript) to clarify this calculation.
* * *
**Response to Comments of Reviewer 2:**

**Major comment #1:** Manuscript needs to document equation illustrating how *Z* is calculated from PSD.

**Response:** An explanation of how *TWC* and *Z* were derived from the PSDs, including addition of Eq. (1) and Eq. (2) that define $TWC_{SD}$ and $Z_{SD}$, has been added on line 27 of page 6. Please see the response to Reviewer 1's major comment #1 for a full description.

**Major comment #2:** The authors have to quantify and discuss S band radar reflectivity factor sensitivity as a function of crystal size.

**Response:** The addition of how *Z* is computed from particle size distributions (Eq. (2) in the revised manuscript) should show that smaller particles do not make as large of contributions to radar reflectivity at S-band as do larger particles given the dependence of radar reflectivity on mass-squared. Further, an example of the cumulative reflectivity distribution function (Fig. (4)) reiterates the point that small particles make small contributions to *Z* (note the y-axis is on a log-scale in the figure).

**Major comment #3:** Don't use B&F for $MMD_{max}$ calculation (Fig 11, etc... and respective arguments in text). B&F has been retrieved for mean chord length size definition and therefore would necessarily overestimate TWC, real $MMD_{max}$, etc...when using PSD in $D_{max}$ definition.

**Response:** The Brown and Francis (BF95) coefficients mentioned in the manuscript were the modified values of Hogan et al. (2012) that correspond to the same definition of particle maximum dimension used in this study. While any set of *m-D* coefficients from previous studies could have been used for comparison purposes, the BF95 coefficients were chosen because they are well cited in the literature. These points have been clarified in lines 28–29 on page 9 (first excerpt below) and lines 14–15 on page 13 (second excerpt below), and text mentioning the BF95 coefficients were modified throughout the remainder of the manuscript to emphasize the modified coefficients were applied in the calculations.

**Text:** The $Z_c(D)$ derived using BF95 coefficients, with the prefactor *a* (=0.002 g cm$^{-1.9}$)

modified following the correction factor of Hogan et al. (2012) applicable for the definition of $D$ used here, is also shown for reference. It is worth noting that the modified BF95 coefficients may reasonably resolve the particle mass for some particle sizes for the PSD depicted in Fig. 4.

**Text:** The $D_{mm}$ is derived using the modified BF95 coefficients to compare among the different flight legs.

**Major comment #4:** Either the manuscript has to derive mathematically how $TWC_{diff}$ (equ 1) and $Z_{diff}$ (equ. 2) can be merged into a sum or likewise into chi-square (equ.3) both to be minimized subsequently (Fig 2). As it stands, we feel that differences (measured, calculated from PSD) of the square root of $Z$ may be a bit comparable to respective differences in $TWC$, in order to merge those two terms. Of course this is scientifically insufficient... The authors may think about sensitivity studies for table 1 data to know which term controls chi-square (equ 3), as a function of different flight legs and how this evolves during individual flight legs.

**Response:** The relative importance of different particle sizes to calculations of bulk parameters is different for $Z$ and $TWC$ ($Z$ is more sensitive to larger particles than is TWC). Thus, depending on the relative concentrations of different sized particles, $Z$ and $TWC$ will be impacted differently depending on the cloud conditions and size distributions present. For the analysis presented in the manuscript, $TWC_{diff}$ has a greater impact on the $\chi^2$ minimization technique some of the time and $Z_{diff}$ some other times. Table 2 has been added to outline the magnitude of $Z_{diff}$ and $TWC_{diff}$ for each flight leg used in the study. $TWC_{diff}$ and $Z_{diff}$ were ultimately not adjusted to be given equal weight in determining the ($a,b$) coefficients for each flight leg as doing so would diminish the utility of using both $TWC$ and $Z$ observations, and limit the ability to diagnose periods when varying size distributions influence $TWC$ and $Z$ differently. Please see the response to Reviewer 1's major comment #4 for further details and how this point has been addressed in the text.

[Figure]

**Major comment #5:** Poisson statistics doesn't take into account systematic uncertainties in measuring *TWC* from Nevzorov and not systematic errors in *Z* measurements. Within a homogeneous cloud segment the statistical counting uncertainty is small, but systematic measurement uncertainty can be of the order of 50% or 100 %... This has to be taken into account instead of Poisson statistics that seems serving to get rid of the uncertainty discussion... ? To resume, the tolerance of EPS has to take into account real uncertainties and not just Poisson statistics / natural variability.

**Response:** The reviewer makes a good point here. The chi-square minimization procedure has been modified to consider measurement uncertainties from the OAPs, Nevzorov probe, and radar. As a result, figures related to the equally plausible surfaces are different for some of the flight legs than those present in the original manuscript. The allowable tolerance $\Delta\chi^2$ for a flight leg now considers the maximum value of the following quantities: natural variability ($\chi^2_{min}$), uncertainty due to Poisson statistics ($\Delta\chi^2_1$), and measurement uncertainties ($\Delta\chi^2_2$). This last term is outlined in Eq. (7) and described on line 22 of page 8. It considers a 50% uncertainty in the PSDs when deriving $TWC_{SD}$ and $Z_{SD}$ (see Heymsfield et al. [2013] and related text in this manuscript), a 2–8% uncertainty in the Nevzorov *TWC* measurements (see Korolev et al. [2012] and related text in this manuscript), and a 1 dB*Z* uncertainty in the radar *Z* measurements (see Krajewski and Ciach [2003] and related text in this manuscript). Fig. 3 has been modified to include distributions of the ratio between $\chi^2_{min}$ and $\Delta\chi^2_2$, and key finding #4 in the conclusions section has been modified accordingly.

**Text:** Estimates of the measurement uncertainty from the OAPs, Nevzorov probe, and ground-based radar also influence the uncertainty in the derived coefficients. The uncertainty due to measurement error $\Delta\chi^2_2$ is defined as [the sum of measurement error from the OAPs, Nevzorov *TWC* probe, and radar]. The terms $TWC_{SD,meas\_min}$, $TWC_{SD,meas\_max}$, $Z_{SD,meas\_min}$, and $Z_{SD,meas\_max}$ represent the minimum and maximum *TWC* or *Z* derived using a 50% uncertainty in the measured *N(D)*. This uncertainty follows Heymsfield et al. (2013) where up to a 50% difference in the number

concentration for particles with $D > 0.1$ mm was determined. Uncertainties in the bulk measurements of *TWC* and *Z* must also be considered in the generation of the uncertainty surfaces with the minimum and maximum possible bulk values represented as $TWC_{meas\_min}$, $TWC_{meas\_max}$, $Z_{meas\_min}$, and $Z_{meas\_max}$. Following Korolev et al. (2013b), it was assumed that there was a 2% uncertainty when Dmax $\leq$ 4 mm and a 8% uncertainty for other periods to address the possibility of particles bouncing out of the cone of the Nevzorov probe. A radar reflectivity uncertainty of 1 dB (Krajewski and Ciach, 2003) is subtracted from or added to the measured *Z* to determine $Z_{meas\_min}$ and $Z_{meas\_max}$.

**Major comment #6:** Recommendation to limit b to 1-3. As a consequence delete everything from pg 7 line 20 to pg 8 line 21, since this discussion finally does not contribute to the study. I don't see the physical meaning to go beyond b=3 of a sphere, mathematically of course one may not care about an interpretation.

**Response:** Although equally plausible solutions that go beyond b $=$ 3 are ultimately not used for the main discussion sections, the existence of these values are interesting from a mathematical standpoint and highlight a major finding from the study that *a* and *b* are highly correlated. Further, provided *a* and *b* are properly correlated there is not anything necessarily unphysical about values of b $>$ 3 (a value of b $>$ 3 does not mean that the *TWC* is greater than that of a sphere over the size range of measured particles provided the a is appropriately chosen—a corollary of this would be a different way in which the density varies with the size of the particles). Therefore, we did not delete the text that the reviewer suggested we delete.

**Major comment #7:** Use PSD number uncertainty of 50% for larger particles and calculate associated $TWC_{SD}$ uncertainty.

**Response:** Please see the response to Reviewer 2's major comment #5 as this point is addressed there.

**Major comment #8:** Likewise $Z_{SD}$ uncertainty from $Z$ equation that you have to present. See comment above.

**Response:** Please see the response to Reviewer 2's major comment #5 as this point is addressed there.

**Major comment #9:** How supercooled water has been quantified? Excluded from analysis? Be aware of the fact that Nevzorov will not correctly quantify *LWC* when *IWC* is more or less dominating *TWC*.

**Response:** We only use ice-phase measurements in our analysis and exclude all liquid-phase and mixed-phase periods from the work. We did this by removing all points from the analysis (rather than attempt to determine the *IWC* in a mixed-phase cloud) if the concentration from the CDP exceeded 10 cm$^{-3}$ for any 1-s period during the 10 second interval. This is addressed in line 18 on page 4.

**Text:** To further constrain the study to periods when clouds were dominated by ice phase hydrometeors such that *TWC* $\approx$ *IWC* and to reduce the impact of liquid phase hydrometeors on the derived *TWC* and *Z*, observations were excluded from the analysis if the concentration from the cloud droplet probe exceeded 10 cm$^{-3}$ at any point during the 10 s interval which usually corresponds to the presence of water (Heymsfield et al., 2011).

**Major comment #10:** Please document average number and mass PSD (additional figure!) for table 1 flight sequences.

**Response:** A figure has been added (Fig. 11 in the revised manuscript) that details the mean *N(D)* and the mean cumulative *M(D)* for each flight leg in this study, and is mentioned in line 11 on page 13 and referenced throughout Sect. 5.3.

**Text:** Figure 11 shows the mean *N(D)* and cumulative mass distribution function *M(D)* using the modified BF95 relationship for each flight leg analyzed in this study.

**Major comment #11:** Nevzorov *TWC* uncertainty: In literature several times has been documented that 2DC+2DP is matching the shallow Nevzorov *TWC*. Likewise for the deep cone in other publications. This illustrates that *a* and *b* coefficients can be adapted to match *TWC* from instruments, but it does not prove that shallow or deep cone Nevzorov collect ice at 100%. May be something between 50-100% for the deep cone. Check literature. Thus, the manuscript has to take into account possible systematic uncertainties of PSD number, Nevzorov TWC, and also measured *Z* impacting the tolerance of EPS and finally guiding recommendations of most likely *a* and *b* coefficients.

**Response:** Please see the response to Reviewer 2's major comment #5 as the Nevzorov *TWC* uncertainty is quantitatively addressed there. The issue of particle bounce out is also acknowledged there.

**Major comment #12:** Figure 9: if the underlying dataset would be statistically a little more representative (which is certainly not the case) wouldn't we expect more organized colours (contour plot gradient) perpendicular to the diagonal line of the grey matrix elements?

**Response:** The reviewer brings up a fair point. Trends in these values may be clearer if computation of the overlap region did not use a fine resolution of (*a,b*) values within the domain described in Sect. 3, or if there was a more statistically representative sample. Additionally, text has been added to lines 15–21 on page 12 to clarify instances where the percentage of overlap between two flight legs may not be the same.

**Text:** Thus, it is possible for the percentage of overlap between two flight legs to be greater when normalized by an equally plausible surface that is smaller in area, and a smaller degree of overlap when normalized by a larger equally plausible surface. It is worth noting that the percentage of overlap does not always follow an organized trend with respect to moving away from the gray diagonal line in the matrix as depicted in the top right corner of Fig. 9a. The lack of organized overlap values in some regions of

the matrix could be influenced by the sensitivity in computing the overlap region over a fine resolution of (*a,b*) values within the domain described in Sec. 3, or perhaps could change in a more organized manner if there was a more statistically representative sample for these calculations to be made.

**Major comment #13:** 20 May flight: these are solely 'low altitude' data with T>-23, whereas for the other two flights used in this study, temperature T<-22. What is the argument to choose the 20 May flight since limited comparison with colder temperature data of two other flights?

**Response:** Inclusion of the 20 May event provided an additional environment for this study as airborne measurements were taken within the trailing stratiform region on this day. While the coldest temperature flight leg (an altitude of 7.9 km) is warmer than the coldest environment from the other two events (8.3–9.1 km in height), there remains a clear temperature dependence on the derived equally plausible surfaces. A brief rationale explaining why these particular events were chosen has been added to lines 21–22 on page 10.

**Text:** These particular events were chosen because of variations in how the complex of storms evolved and the location of in situ measurements relative to the convective system.

**Minor comment #1:** Pg 1, line 18: It does not make sense to establish *m-D* relation in mixed phase clouds. We know *m-D* for water droplets. You should exclude all mixed phase sequences from data!

**Response:** We agree with the reviewer and tried to better clarify what we did. Periods where *LWC* was suspected to represent a notable component of the *TWC* were removed from the analysis, with stringent criteria developed to avoid notable contributions from supercooled water. Please see how this is detailed in the response to Reviewer 2's major comment #9. Further, explicit reference to mixed phase clouds has

been removed from the last sentence in the abstract (line 18 on page 1) so that any confusion is avoided.

**Text:** These findings show the importance of representing the variability in a,b coefficients for numerical modeling and remote sensing studies rather than assuming fixed values, as well as the need to further explore how these surfaces depend on environmental conditions in clouds containing ice hydrometeors.

**Minor comment #2:** Figure 1 is not thoroughly documented with literature references and thus not reproducible.

**Response:** A supplemental table of these references (separate from the manuscript) has been added that includes the coefficients themselves, the environment and method in which they were derived, and pertinent notes. This table is mentioned in line 10 on page 2.

**Minor comment #3:** Pg 4, line 14: redo analysis with CDP droplet probe not exceeding 10 cm$^{-3}$ on a 1 s basis may be a more suitable approach.

**Response:** The analysis has been redone using a threshold based on the 1 HZ CDP data, and is mentioned in line 21 on page 4 and also in the response to Reviewer 2's major comment #9.

**Minor comment #4:** Pg 4, line 18 ff: Assuming that S band radar reflectivity is not significant for sub-millimetric particles, what is the *TWC* percentile at 1mm of the cumulative mass PSD ? Please don't use B&F but for example Heymsfield 2010, etc.. retrieved for $D_{max}$ definition.

**Response:** Figs. 11d-f in the revised manuscript provide the information necessary to determine estimates of the *TWC* percentile at 1 mm for each flight leg. As mentioned in response to the previous comments, clarifying that the BF95 coefficients were modified

following Hogan et al. (2012) should address the reviewer's concern of an appropriate *m-D* relation for the derived cumulative *M(D)*.

**Minor comment #5:** Pg 8, line 14: 85 cm?

**Response:** The largest possible *D* that has particle masses (from the *a* and *b* at the 95th percentile in Fig. 4) less than those of spherical particles with a density of solid ice for the same *D* is computed by rearranging the equation $\rho_{ice} = (aD^b)/(\pi/6 \times D^3)$. This calculation is intended to merely reiterate the point in the previous sentence of the manuscript that "bulk variables such as *Z* derived using $b > 3$ are physically plausible" only when "the coefficients are applied over the range of particle sizes observed...". After addressing Reviewer 2's major comment #5, the *a* and *b* at the 95th percentile now permit a *D* as large as 3.83 cm.

**Minor comment #6:** Pg 9, line 26: Use *m-D* relation, for mass content estimation, other than BF95 for $D_{max}$ size definition.

**Response:** Please see the response to Reviewer 2's major comment #3 as this point is fully addressed there.

**Minor comment #7:** Pg 11, line 5; Fig 11: Idem

**Response:** Please see the response to Reviewer 2's major comment #3 as this point is fully addressed there.

**Minor comment #8:** Pg 15, line 15 idem

**Response:** Please see the response to Reviewer 2's major comment #3 as this point is fully addressed there.

**Minor comment #9:** Pg 11, line 12; You mean just Fig 11 in this sentence?

**Response:** Figures 10–12 are all ordered the same as in Fig. 6, with "instances of multiple legs having the same average temperature shown in chronological order."

**Minor comment #10:** Pg 11, equation 6: How sphericity results deviate when comparing two instruments of different pixel resolution having sampled the same paricle? And how the averaging of sphericity has been performed over all crystals of a crystal population? For OAP 2D images the sphericity is size dependent, partly due to bias of 2D projections sorted into size classes, with larger particles having smaller 2D image sphericity than smaller particles.

**Response:** It is true that the probe resolution can cause differences in the computed sphericity, particularly at smaller particle sizes. However, the diode resolution is much less sensitive to sphericity as larger sizes are approached. Simple calculations of sphericity were conducted at 1.0 mm (the size at which the $N(D)$ was found to agree within 5% between the 2D-C and HVPS during MC3E) that used a synthetically-generated spherical particle based on the probe resolution of the 2D-C and HVPS. At 1.0 mm, the percent difference in maximum sphericity between the HVPS and 2D-C is only 4.8%. The maximum sphericity for a circular particle was also computed using the HVPS resolution for larger sizes and found to vary by 6.8% between 1 mm and 10 mm. As such, the effect of diode resolution and particle size suggest that sphericity can be used as a quantitative comparison between flight legs for the points addressed in the manuscript.

**Minor comment #11:** Pg 12 line 16: Why applying factor 0.6 in reference volume for retrieving effective density, impacting TWC???

**Response:** Since most of the observed ice hydrometeors are non-spherical, the volume used in deriving effective density here applies the assumption that particles are enclosed by an oblate spheroid with a typical aspect ratio of 0.6 following observations from previous studies (e.g., Hogan et al., 2012). This point has been clarified in line 28

on page 14.

[revised manuscript text omitted]